# ArenaRL: Scaling RL for Open-Ended Agents via Tournament-based Relative Ranking

Qiang Zhang [* 1]  Boli Chen [2]  Fanrui Zhang [* 1 3]  Ruixue Ding [2]  Shihang Wang [2]  Qiuchen Wang [1]
Yinfeng Huang [4]  Haonan Zhang [4]  Rongxiang Zhu [4]  Xin Li [4]  Houquan Zhou [2]  Pengjun Xie [2]  Kaipeng Zhang [3]
Jingren Zhou [2]  Jiawei Liu [1]

## Abstract

Reinforcement learning (RL) has advanced LLM agents on verifiable tasks but remains challenging for open-ended tasks with vast solution spaces (*e.g.*, complex travel planning). Lacking objective ground truth, current RL algorithms rely on reward models assigning scalar scores to individual responses. We contend such pointwise scoring induces discrimination collapse: reward model fails to distinguish subtle advantages among trajectories, compressing intra-group rewards into a narrow range. This drowns effective reward signals in reward model noise, causing optimization stagnation. To address this, we propose ArenaRL, a reinforcement learning paradigm shifting from pointwise scalar scoring to intra-group relative ranking. ArenaRL introduces a process-aware pairwise evaluation with multi-level rubrics for fine-grained relative scoring. Meanwhile, we construct an intra-group adversarial arena and devise a tournament-based ranking scheme to obtain stable advantage signals. ArenaRL achieves high-precision advantage estimation with only $O(N)$ computational complexity, striking a favourable balance between efficiency and accuracy. Furthermore, to address the lack of full-cycle benchmarks for open-ended agents, we introduce two high-quality benchmarks: Open-Travel and Open-DeepResearch, encompassing full training and multi-dimensional evaluation pipelines. Extensive experiments across three open-ended tasks validate the effectiveness of ArenaRL.[1]

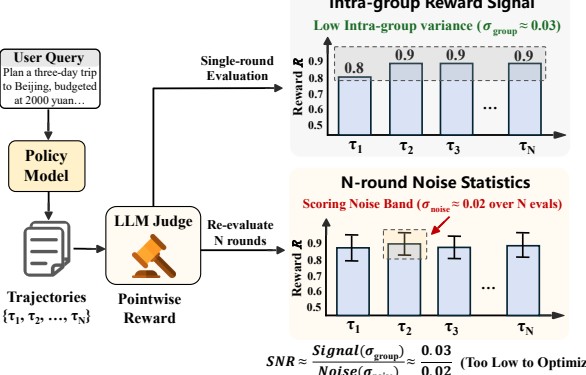

*Figure 1.* **Illustration of discriminative collapse in pointwise evaluation:** We analyse the trajectory rewards for a query example during RL training under two settings. **(1) Intra-group Reward Signal**, representing the pointwise reward and intra-group variance ($\sigma_{\text{group}}$) for a trajectory group; and **(2) N-round Noise Statistics**, depicting the average reward and scoring noise band across $N$ independent evaluation repetitions to estimate noise variance ($\sigma_{\text{noise}}$). We observe that the $\sigma_{\text{noise}}$ is substantial and comparable to $\sigma_{\text{group}}$. This yields an extremely low signal-to-noise ratio (SNR), obscuring genuine advantages and hindering effective RL optimization.

---

[*]Equal contribution  [1]MoE Key Laboratory of Brain-inspired Intelligent Perception and Cognition, University of Science and Technology of China [2]Tongyi Lab, Alibaba Group [3]Shanghai Innovation Institute [4]Amap, Alibaba Group. Correspondence to: Jiawei Liu <jwliu6@ustc.edu.cn>.

*Proceedings of the $43^{rd}$ International Conference on Machine Learning*, Seoul, South Korea. PMLR 306, 2026. Copyright 2026 by the author(s).

[1]Code: https://github.com/Alibaba-NLP/qqr

## 1. Introduction

The evolution of large language models (LLMs) into autonomous agents marks a paradigm shift in artificial intelligence from passive question answering to active problem solving. By integrating long-horizon planning and tool use, LLM agents have shown great potential in handling complex tasks (Yao et al., 2022b; Team et al., 2025). In this progression, reinforcement learning (RL) has played a pivotal role, particularly in deterministic tasks such as mathematical reasoning and code generation, where ground-truth labels provide explicit reward signals for optimization (Li et al., 2025a). However, extending RL to open-ended agent tasks in real-world scenarios, such as personalized travel planning or in-depth industry analysis, poses fundamental challenges (Li et al., 2025b). In these domains, the solution spaces are vast and unstructured, and the notion of correctness is subjective and multi-dimensional, rendering traditional RL algorithms inapplicable (Jia et al., 2025).

Mainstream RL approaches (*e.g.*, GRPO (Shao et al., 2024), GSPO (Zheng et al., 2025)) typically assume access to a verifiable reward function that provides accurate feedback. However, this assumption often fails in open-ended settings where no objective ground truth is available. To address reward acquisition in such scenarios, recent works adopt the LLM-as-Judge paradigm, assigning pointwise scalar scores to model outputs (Viswanathan et al.; Huang et al., 2025; Liu et al., 2025). We identify that this paradigm induces a severe issue we term discriminative collapse, as shown in Figure 1. As the policy is refined, generated trajectories become increasingly similar in distribution, and the pointwise scoring scheme compresses rewards within the same group into a narrow band (*e.g.*, 0.8–0.9 on a scale of 1), making them effectively indistinguishable. Moreover, due to inherent noise in the LLM judge, such as decoding randomness (Arias et al., 2025) and length preferences (Hu et al.), the reward outcomes exhibit a certain degree of unreliability, with a low signal-to-noise ratio between the reward signal and the interfering noise. In this regime, the pointwise evaluation fails to reliably identify the truly superior samples. And the RL optimization process is driven more by spurious noise than by meaningful task-specific signals, leading performance to stagnate or even deteriorate.

To address this, we draw inspiration from decision theory, where pairwise preference judgments are known to be more stable than pointwise quantitative assessments (Fürnkranz & Hüllermeier, 2010; Rafailov et al., 2023), and advocate a paradigm shift from pointwise scalar scoring to intra-group relative ranking. To this end, we propose ArenaRL, an online policy optimization framework grounded in an adversarial arena. ArenaRL abandons unstable scalar rewards in favor of constructing relative rankings over trajectories within each generated group. To ensure both depth and fairness in evaluation, we introduce a process-aware pairwise evaluation mechanism that not only compares the reliability of the outcome, but also scrutinizes the logical coherence of the chain-of-thought and the effectiveness of tool invocations along the trajectory.

A central bottleneck in scaling pairwise preference optimization to open-ended agent tasks lies in the high computational cost. While exhaustive comparisons yield accurate rankings, the resulting $O(N^2)$ complexity is intractable for online training. To explore the trade-off between ranking fidelity and training efficiency, we designed five tournament topologies, ranging from exhaustive round-robin to single and double elimination formats. Our empirical analysis reveals a critical challenge: standard single and double tournaments are highly sensitive to the initial pairing. Random matching often leads to the premature collision and elimination of high-quality trajectories, degrading global ranking accuracy. To mitigate this, we propose a seeded single-elimination mechanism. This approach utilizes a greedy-decoded trajec-

tory as a quality anchor to establish a low-bias prior. These initial estimates seed the tournament, which then follows a structured binary-tree competition. Our results indicate that this design reduces computational complexity to linear $O(N)$ while robustly preserving the fidelity of intra-group rankings, achieving an optimal balance between training efficiency and advantage estimation accuracy.

Furthermore, to address the lack of full-cycle benchmarks for open-ended agents, we introduce two comprehensive benchmarks: (1) Open-Travel, which focuses on evaluating agents' long-horizon planning capabilities under multiple hard constraints such as budget and spatiotemporal windows; and (2) Open-DeepResearch, which centers on assessing agents' abilities in autonomous information retrieval and report generation in realistic internet environments. Unlike traditional benchmarks that only provide a test set (He et al., 2025; Du et al., 2025; Coelho et al., 2025), the two proposed benchmarks offer a complete pipeline from supervised fine-tuning $\rightarrow$ RL training $\rightarrow$ multi-dimensional automated evaluation, establishing a reproducible infrastructure for the community. Given the shared characteristics of open-ended tasks, we further extend our experiments to standard open-ended writing tasks (Wang et al., 2025a) and conduct systematic evaluations on three public benchmarks. Extensive experiments demonstrate that ArenaRL significantly outperforms strong baselines across travel planning, deep research, and open-ended writing tasks.

In summary, our major contributions are as follows: **(1)** We identify and formalize the problem of discriminative collapse in open-ended tasks, and propose ArenaRL, which replaces unstable pointwise scalar rewards with a tournament-based relative ranking mechanism to enable robust policy optimization. **(2)** To address the high computational cost of pairwise comparisons, we design and validate a seeded single-elimination tournament topology that achieves high-accuracy advantage estimation with only $O(N)$ complexity. **(3)** We construct the Open-Travel and Open-DeepResearch benchmarks with full training pipelines, filling a critical gap in evaluating the full lifecycle of open-ended agents.

## 2. Related Work

**Open-Ended Agent Benchmark.** The rapid development of LLMs has given rise to autonomous agents that interact with external environments and solve complex tasks (Guo et al., 2025). While established benchmarks like WebShop (Yao et al., 2022a), Mind2Web (Deng et al., 2023), and SWE-bench (Jimenez et al.) focus on deterministic settings with verifiable feedback, they often overlook open-ended, real-world applications like travel planning (Ning et al., 2025) and industrial research (Li et al., 2025b). In these domains, the absence of a unique gold solution requires agents to balance multi-faceted constraints and user utility. Cru-

cially, the community lacks a systematic training–evaluation infrastructure for such agents. Existing benchmarks (*e.g.*, VitaBench (He et al., 2025), DeepResearchBench (Du et al., 2025)) are primarily static test sets designed for post-hoc evaluation. To address this, we introduce Open-Travel and Open-DeepResearch. Unlike traditional benchmarks, they offer an integrated pipeline covering SFT, RL exploration, and multi-dimensional evaluation to systematically study agent capabilities.

**Reinforcement Learning with LLMs.** Reinforcement learning (RL) has emerged as a powerful paradigm for aligning LLMs with complex objectives (Zhang et al., 2025; Gao et al., 2025). In domains with clear ground truth, algorithms like GRPO (Shao et al., 2024) and DAPO (Yu et al., 2025) have achieved significant success. However, applying RL to open-ended tasks lacking objective outcomes remains challenging. Since correctness is subjective, mainstream methods typically adopt the LLM-as-Judge paradigm, optimizing agents via pointwise scalar scores (Viswanathan et al.; Huang et al., 2025). Yet this approach struggles to distinguish subtle advantages between trajectories within a group, leading to discriminative collapse. Recent studies have begun exploring comparison-based evaluation: Writing-Zero (Jia et al., 2025) assigns binary advantages by comparing responses against random references, while Pref-GRPO (Wang et al., 2025b) derives reward from win rates via exhaustive pairwise comparison. Despite these advances, this mechanism remains unexplored for long-horizon agent tasks. Moreover, Writing-Zero offers only coarse-grained binary guidance, and exhaustive comparisons with $O(N^2)$ complexity are cost-prohibitive for online training. ArenaRL addresses these limitations with a sparse tournament-based ranking, achieving high-fidelity advantage estimation with linear $O(N)$ complexity.

## 3. Preliminary

**Task Definition.** We formulate the open-ended agentic task as a conditional trajectory generation problem. Given a query $x$ from task distribution $\mathcal{D}$ and a tool set $T$, the agent policy $\pi_\theta$ generates a multi-step trajectory $\tau$. Formally, $\tau$ is an interleaved sequence comprising chain-of-thought $z_k$, tool invocations $a_k \in T$, environmental feedback $o_k$, and a final answer $y$:

$$\tau = [z_1, a_1, o_1, \ldots, z_k, a_k, o_k, \ldots, z_K, a_K, o_K, y] \quad (1)$$

The optimization objective is to maximize a reward signal while penalizing deviation from a reference policy $\pi_{\text{ref}}$:

$$\mathcal{L}(\theta) = \mathbb{E}_{x \sim \mathcal{D}, \tau \sim \pi_\theta(\cdot|x;T)} \left[ r_\phi(x, \tau) - \beta \mathbb{D}_{\text{KL}} \big( \pi_\theta(\cdot|x) \, \| \, \pi_{\text{ref}}(\cdot|x) \big) \right] \quad (2)$$

where $r_\phi(x, \tau)$ denotes the reward signal for trajectory $\tau$.

**Reward Modeling and SNR Analysis.** In practice, the reward $r_\phi$ is often provided by an LLM-as-judge, which we model as $r_\phi = Q(x, \tau) + \epsilon$. Here, $Q(x, \tau)$ represents the true latent quality of the trajectory, and $\epsilon \sim \mathcal{N}(0, \sigma_{noise}^2)$ denotes the inherent scoring noise of the judge. Under this framework, the total variance of the observed reward is decomposed as:

$$Var(r_\phi) = Var(Q) + Var(\epsilon) = \sigma_{group}^2 + \sigma_{noise}^2 \quad (3)$$

where $\sigma_{group}$ denotes the quality variance within a trajectory group. We define the Signal-to-Noise Ratio (SNR) as:

$$SNR = \frac{\sigma_{group}}{\sigma_{noise}} \quad (4)$$

As policy $\pi_\theta$ is optimized, trajectories within a group $\mathcal{G}$ become increasingly homogeneous, leading to $\sigma_{group} \to 0$. Take Figure 1 as example, when $SNR \approx 1$, the reward signal is overwhelmed by noise, causing the gradient estimate to lose direction and resulting in optimization stagnation.

**Process-Aware Pairwise Evaluation.** To mitigate the instability of pointwise scalar scoring, we adopt a pairwise evaluation paradigm via an Arena Judge $\mathcal{J}$. Given a query $x$ and two candidate trajectories $(\tau_i, \tau_j)$, the judge $\mathcal{J}$ evaluates them jointly to output their respective quality scores.

Specifically, the judge $\mathcal{J}$ processes three components: (1) the user query $x$; (2) the core context of trajectories $\tau_i$ and $\tau_j$ (containing the chain-of-thought (CoT) $z_k$ and tool invocations $a_k$ for each step, and the final answer $y$); and (3) a comprehensive process-aware rubric $u$. This rubric enforces fine-grained scrutiny of logical consistency in the CoT, the precision of tool calls, and the reliability of the final answer. This ensures that the optimization signal reinforces the agent's intrinsic reasoning capabilities rather than merely overfitting to surface-level features of the final answer. Detailed prompts are in Appendix E.

To counter the positional bias in LLM judges (Wu et al., 2025b), we employ a bidirectional scoring protocol. We conduct two independent evaluations by swapping the trajectory order and aggregating the results:

$$(s_i, s_j) = \mathcal{J}(u, x, \tau_i, \tau_j) + \mathcal{J}(u, x, \tau_j, \tau_i) \quad (5)$$

where $s_i$ and $s_j$ represent the final debiased quality scores.

## 4. Methodology: ArenaRL

In this section, we detail the proposed ArenaRL (Figure 2), which redefines reward modeling as relative quality ranking within a trajectory group $\mathcal{G} = \{\tau_1, \ldots, \tau_N\}$ from policy $\pi_\theta$. By constructing a dynamic arena, ArenaRL derives robust advantage signals. We systematically design five tournament topologies to identify the optimal trade-off between computational cost and ranking fidelity.

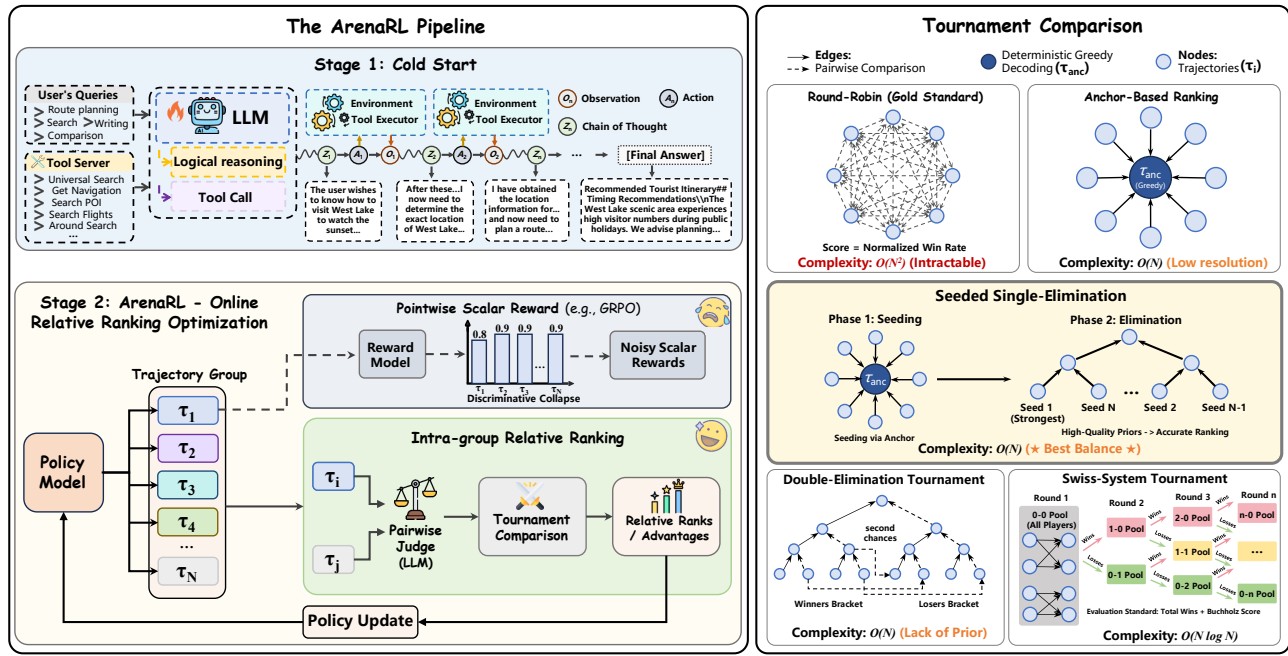

*Figure 2.* The overall of the proposed ArenaRL algorithm. ArenaRL replaces conventional pointwise scalar reward paradigm with intra-group relative ranking and designs five distinct tournament topologies to optimally balance computational cost and ranking fidelity.

## 4.1. Round-Robin Tournament

In this scheme, each trajectory $\tau_i$ competes against all other $N-1$ trajectories via our process-aware pairwise evaluation. The final score is defined as the normalized win rate:

$$w_i = \frac{1}{N-1} \sum_{j \neq i} \mathbb{I}(s_i > s_j) \quad (6)$$

where $\mathbb{I}(\cdot)$ denotes the indicator function, which equals 1 if the score $s_i$ is greater than $s_j$, and 0 otherwise. Group rankings are then derived by sorting $w_i$ in descending order. Despite providing theoretically unbiased intra-group rankings, its quadratic complexity $O(N^2)$ is prohibitive for large-scale online training. Consequently, we employ Round-Robin primarily as a "gold standard" to benchmark the fidelity of more efficient topologies.

## 4.2. Seeded Single-Elimination

To balance computational efficiency and ranking fidelity, we propose Seeded Single-Elimination. Unlike naive elimination formats that suffer from high ranking bias due to stochastic pairings, this approach utilizes a deterministic anchor to establish a robust prior before hierarchical competition. The procedure consists of three phases:

**(1) Anchor-Based Prior Seeding.** We define the trajectory group as $\mathcal{G} = \{\tau_{anc}\} \cup \{\tau_1, \ldots, \tau_{N-1}\}$, where $\tau_{anc}$ is the quality anchor from greedy decoding, and the remaining $N-1$ trajectories constitute high-entropy sampled exploratory samples. We first perform $N-1$ pairwise comparisons of all exploratory samples against $\tau_{anc}$ to derive an initial score set $\mathcal{S}_{init} = [s_{anc}, s_2, \ldots, s_N]$:

$$(s_{anc}^i, s_i) = \mathcal{J}(u, x, \tau_{anc}, \tau_i), \quad \forall \tau_i \in \mathcal{G} \setminus \{\tau_{anc}\} \quad (7)$$

The anchor's quality is estimated as $s_{anc} = \frac{1}{N-1} \sum s_{anc}^i$. By sorting $\mathcal{G}$ descendingly based on $\mathcal{S}_{init}$, we obtain an ordered seed sequence, providing a low-bias estimate of the global quality distribution.

**(2) Hierarchical Competitive Refinement.** We employ serpentine grouping scheme to construct the initial bracket $\mathcal{B}_1$, pairing top-seeded trajectories with bottom-seeded ones (*e.g.*, Seed $k$ vs. $N-k+1$), thereby preventing premature collisions between top-tier trajectories.

In each competition round $m$, paired candidates $(\tau_a, \tau_b) \in \mathcal{B}_m$ are evaluated by the arena judge $\mathcal{J}$, yielding corresponding quality scores $(s_a, s_b)$. The winning trajectory ascends to the next tier $\mathcal{B}_{m+1}$ to compete in the subsequent round, while the losing trajectory is archived into a ranking tier $\mathcal{T}_m$. Throughout the tournament, we maintain a cumulative quality value for each trajectory, denoted as $V[\tau_i]$:

$$V[\tau_i] \leftarrow V[\tau_i] + s_{\tau_i} \quad (8)$$

This value accumulates the quality scores obtained from all competitions participated in by each trajectory.

**Rank Assignment.** Upon tournament completion, the final ranking $\text{Rank}_{\tau_i}$ is determined by two factors: (1) Survival

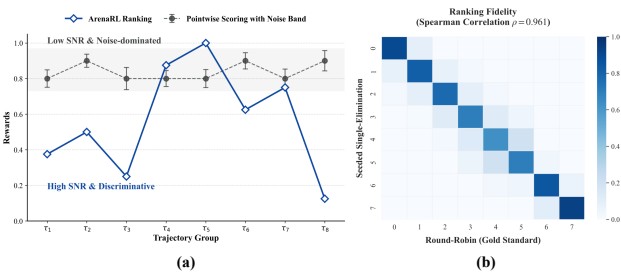

*Figure 3.* **(a)** Comparison of reward distributions between point-wise scoring and ArenaRL. **(b)** Ranking consistency between the Seeded Single-Elimination and Round-Robin Tournament across 50 trajectory groups.

Depth: Trajectories are ordered by the round in which they were eliminated (*i.e.*, $\mathcal{T}_{m+1} \succ \mathcal{T}_m$). (2) Intra-tier ranking refinement: For trajectories eliminated within the same tier, to resolve ties, a finer distinction is made by comparing the magnitude of their quality values $V[\tau]$.

Beyond the above two schemes, we explored three other topologies: Anchor-based ranking, Double-elimination tournament and Swiss system. Details refer to Appendix B.

### 4.3. Ranking-Based Policy Optimization

Irrespective of the underlying tournament topology, ArenaRL produces a relative ranking $\text{Rank}_{\tau_i} \in \{0, \ldots, N-1\}$ for each trajectory in the group, where 0 denotes the highest rank. To enable stable optimization, we convert these discrete ranks into normalized advantage signals. We first map the ranks to quantile-based rewards:

$$r_i = 1 - \frac{\text{Rank}_{\tau_i}}{N - 1}. \tag{9}$$

We then compute the advantage $A_i$ for each trajectory:

$$A_i = \frac{r_i - \mu_r}{\sigma_r + \epsilon}, \tag{10}$$

where $\mu_r$ and $\sigma_r$ denote the mean and standard deviation of the rank-based rewards $\{r_1, \ldots, r_N\}$, respectively.

Finally, we optimize the policy by maximizing the following objective function:

$$\max_{\pi_\theta} \ \mathbb{E}_{x \sim \mathcal{D}, \tau \sim \pi_\theta(\cdot|x;T)} \Big[ r_\phi(x, \tau) - \beta \mathbb{D}_{\text{KL}} \big( \pi_\theta(\tau|x) \parallel \pi_{\text{ref}}(\tau|x) \big) \Big] \tag{11}$$

### 4.4. Empirical Analysis of the Ranking Paradigm

To justify the proposed intra-group ranking paradigm, we analyze its discriminative power and ranking fidelity, as illustrated in Figure 3:

**Eliminating Discriminative Collapse.** As shown in Figure 3 (a), the pointwise scoring suffers from a severe discriminative collapse. Subtle quality differences between trajectories

within the same group cause rewards to be compressed into a narrow interval. This compression, exacerbated by inherent reward model noise, results in a low signal-to-noise ratio (SNR). In contrast, ArenaRL maps these nuances into high-resolution discrete ranking rewards. By naturally filtering out absolute scoring fluctuations and focusing on relative advantages, ArenaRL significantly enhances the SNR.

**Fidelity of the Seeded Single-Elimination.** Figure 3 (b) evaluates the ranking consistency between the Seeded Single-Elimination scheme and the Round-Robin tournament. The results exhibit a strong diagonal concentration with a Spearman correlation of $\rho = 0.961$. This confirms that the Seeded Single-Elimination maintains near-perfect ranking fidelity while reducing the computational complexity from $O(N^2)$ to linear $O(N)$.

## 5. Benchmarking Open-Ended Agency

To systematically evaluate open-ended agent capabilities, we introduce Open-Travel and Open-DeepResearch, two comprehensive benchmarks derived from real-world application scenarios. These benchmarks are constructed via a rigorous three-stage pipeline:

**Benchmark Data Collection.** We collected anonymized query samples from public web platforms, which were then curated and refined by five experts to yield a high-quality query dataset. From this pool, a test set was sampled, with complete trajectories generated by the Claude-Sonnet-4 (Anthropic, 2025). These will serve as baseline trajectories for final pairwise comparisons and win rate assessments.

**Large-Scale Training Data Construction.** To expand the training distribution, we employed diverse LLMs (*e.g.*, Qwen3 (Yang et al., 2025), GPT-4o (Achiam et al., 2023)) as "query generators" to synthesize multi-scenario open-ended queries based on the seed queries from the first phase. For a portion of these queries, we constructed synthetic trajectories using Qwen3-Max and Claude-Sonnet-4 to support supervised fine-tuning. The remaining queries were reserved as unlabeled data for RL exploration.

**Trajectory Quality Control.** We implemented a rule-augmented LLM inspection module to filter synthetic trajectories. This module evaluates tool-use effectiveness, conversational accuracy and logical consistency, iteratively rewriting non-compliant samples to ensure data integrity.

### 5.1. Domains and Statistics

Using the above pipeline, we modularly construct two distinct domains: Open-Travel and Open-DeepResearch, to evaluate agentic reasoning and planning.

**Open-Travel.** This domain requires the agent to execute five types of itinerary planning subtasks, emphasizing multi-

*Table 1.* Statistics of the constructed Open-Travel and Open-DeepResearch benchmarks.

| Dataset | Training Set | | Test Set | Language | Avg. Turns |
| --- | --- | --- | --- | --- | --- |
| | SFT | RL | | | |
| Open-Travel | 2,600 | 1,626 | 250 | Chinese | 7.81 |
| Open-DeepResearch | 2,662 | 2,216 | 100 | Chinese / English | 9.93 |
| **Total** | **5,262** | **3,842** | **350** | Chinese / English | – |

constraint reasoning, tool coordination, and personalized preference alignment. Specifically, the subtasks include: (1) *Direction*: multi-waypoint routing, (2) *1-Day*: single-day trip planning, (3) *Compare*: transportation mode analysis, (4) *Search*: Nearby point-of-interest (POI) discovery, and (5) *M-Day*: complex multi-day itinerary planning.

**Open-DeepResearch.** This domain evaluates the agent's proficiency in multi-turn web search and information synthesis for open-ended response generation. Tasks are categorized into three areas: (1) drafting technical documentation, (2) formulating research plans, and (3) elucidating complex concepts for general accessibility.

**Dataset Statistics.** As shown in Table 1, the benchmarks exhibit diverse styles and reasonable scale for evaluating open-ended agentic capabilities. The training set contains 2,600 SFT and 1,626 RL samples for Open-Travel, alongside 2,662 SFT and 2,216 RL samples for Open-DeepResearch. The test sets, reserved for leaderboard evaluation, consist of 250 and 100 manually validated samples respectively, ensuring high standards of clarity and task difficulty. Both benchmarks involve long-horizon interactions, with average dialogue turns of 7.81 and 9.93, highlighting the multi-step reasoning and long-term planning challenges of open-ended agent tasks.

### 5.2. Evaluation

We adopt the LLM-as-a-judge paradigm, employing Qwen3-Max and Claude-Sonnet-4 models as dual judges to evaluate both reasoning trajectories and final answers.

For Open-Travel, we pairwise compare the trajectories of the model under test against baseline trajectories from the benchmark. Each judge scores the comparison based on multiple rubrics (*e.g.*, correctness, consistency with reasoning) and outputs a preference outcome. For each rubric, we compute the win rate as the proportion of non-tied cases where the model under test is preferred, and average the two judges' results as the final metric for each subtask.

For Open-DeepResearch, we employ the same pairwise evaluation protocol as for Open-Travel. Owing to potential context overflow, we additionally track the valid generation rate (Val., %), the proportion of test instances with a valid final answer. Under valid generation conditions, we compute per-rubric win rates against the baseline and average

these values to derive the final performance score.

## 6. Experiments

To comprehensively evaluate the effectiveness of ArenaRL, we first report the performance of the five tournament topologies introduced in Section 4, justifying our selection of the Seeded Single-Elimination scheme. Building upon this optimal topology, we benchmark ArenaRL against strong baselines on the Open-Travel and Open-DeepResearch datasets. Furthermore, recognizing the shared characteristics of open-ended tasks, we extend our evaluation to open-ended writing tasks using three public benchmarks.

### 6.1. Experimental Settings

**Baselines.** We evaluate ArenaRL against two categories of baselines. **(1)** Four closed-source models: GPT-4o (Achiam et al., 2023), Grok-4 (xAI, 2025), Gemini-2.5-pro (Team et al., 2023), and Claude-3.7-Sonnet (Anthropic, 2023). (2) Three representative RL algorithms: GRPO (Shao et al., 2024), GSPO (Zheng et al., 2025) and Reinforce++ (Hu, 2025), trained under a standard LLM-as-judge setup with pointwise rewards. To maintain fairness, these RL algorithms utilize the exact same judge models and evaluation rubrics as ArenaRL, and only evaluate the answer portion. The implementation details are provided in Appendix A.1.

**Evaluation Metrics.** For open-ended agent tasks, we perform pairwise evaluations against the baseline trajectories in the benchmark using the evaluation approach defined in Section 5, and compute the corresponding win rates. For open-ended writing, we adopt three complementary benchmarks for a comprehensive assessment: WritingBench (Wu et al., 2025a), HelloBench (Que et al., 2024), and LongBench-write (Bai et al.). And following established protocols (Wang et al., 2025a), we employ the LLM-as-judge approach to score the generative quality of different models. The evaluation details are provided in the Appendix A.2.

### 6.2. Tournament Topology Analysis

Table 2 systematically compares different tournament topologies under a unified RL configuration (group size $N = 8$, groups $K = 8$). The results identify Seeded Single-Elimination as the optimal trade-off between efficiency and performance. It achieves an average win rate of 32.5%, comparable to the "gold standard" Round-Robin (32.9%), while maintaining linear complexity $O(N)$. Conversely, Swiss Round and Double-Elimination formats underperform, suffering from either weak initial priors or insufficient comparison depth. These findings suggest that anchor-based seeding effectively shields high-quality candidates from stochastic pairing fluctuations in early stages, ensuring superior ranking accuracy. Consequently, we adopt Seeded

*Table 2.* Performance comparison of the five tournament topologies on the Open-Travel benchmark.

| Topology | Comparison Cost | Open-Travel | | | | | Mean |
|---|---|---|---|---|---|---|---|
| | | Direction | Search | Compare | 1-Day | M-Day | |
| SFT | - | 10.6 | 29.7 | 14.1 | 20.4 | 7.1 | 16.4 |
| Anchor-Based Ranking | $N-1$ | 18.0 | 41.3 | 30.9 | 31.1 | 17.6 | 27.8 |
| Swiss-System | $N \log N$ | 20.9 | 43.0 | 27.9 | 38.6 | 11.1 | 28.3 |
| Double-Elimination | $2N-2$ | 12.6 | 52.4 | **33.7** | **39.9** | 12.3 | 30.2 |
| **Seeded Single-Elimination** | $2N-2$ | 16.9 | **69.9** | 22.9 | 34.9 | 18.1 | 32.5 |
| *Round-Robin (Upper Bound)* | $N(N-1)/2$ | **23.3** | 66.3 | 23.6 | 32.1 | **19.0** | **32.9** |

*Table 3.* Performance comparison on Open-Travel and Open-DeepResearch benchmarks.

| Method | Open-Travel | | | | | | Open-DeepResearch | | | | | | | |
|---|---|---|---|---|---|---|---|---|---|---|---|---|---|---|
| | Direction | Search | Compare | 1-Day | M-Day | **Mean** | Frm. | Tool. | Cov. | Rel. | Acc. | Dep. | Cla. | **Mean (Val. %)** |
| *Closed-source Models* | | | | | | | | | | | | | | |
| GPT-4o | 2.4 | 5.0 | 3.1 | 1.6 | 0.7 | 2.6 | 5.1 | 24.4 | 21.0 | 9.1 | 12.5 | 2.3 | 10.8 | 12.2 (88.0) |
| Grok-4 | 17.0 | 21.3 | 9.7 | 24.7 | 11.3 | 16.8 | 33.7 | 36.8 | 43.4 | 36.8 | 39.2 | 36.1 | 17.5 | 34.8 (83.0) |
| Gemini-2.5-pro | 8.6 | 12.5 | 7.4 | 11.9 | 12.4 | 10.6 | 15.8 | 19.0 | 17.9 | 32.6 | 28.3 | 45.7 | 38.6 | 28.3 (92.0) |
| Claude-3.7-Sonnet | 18.6 | 59.6 | 14.7 | 43.6 | 21.3 | 31.6 | 10.1 | 13.5 | 22.5 | 23.6 | 19.7 | 27.0 | 17.4 | 19.1 (89.0) |
| *Backbone Model: Qwen3-4B-Base* | | | | | | | | | | | | | | |
| SFT | 8.2 | 20.1 | 8.3 | 11.6 | 6.7 | 11.0 | 10.3 | 5.2 | 8.6 | 5.2 | 8.6 | 10.3 | 8.6 | 8.1 (29.0) |
| GRPO | 9.1 | 22.4 | 8.0 | 12.3 | 6.2 | 11.6 | 16.8 | 11.2 | 14.2 | 8.9 | 13.5 | 16.8 | 9.5 | 14.6 (21.0) |
| GSPO | 8.7 | 18.5 | 8.5 | 11.9 | 7.7 | 11.0 | 15.6 | 12.5 | 16.8 | 10.2 | 12.5 | 15.6 | 10.1 | 13.3 (18.0) |
| Reinforce++ | 9.4 | 21.5 | 9.1 | 10.2 | 6.5 | 11.3 | 19.2 | 8.4 | 14.7 | 12.3 | 14.0 | 17.5 | 8.6 | 15.1 (24.0) |
| **ArenaRL** | **17.0** | **45.1** | **24.9** | **28.1** | **16.6** | **26.3** | **43.3** | **68.0** | **69.6** | **30.4** | **35.6** | **25.3** | **34.0** | **43.7 (96.0)** |
| *Backbone Model: Qwen3-8B-Base* | | | | | | | | | | | | | | |
| SFT | 10.6 | 29.7 | 14.1 | 20.4 | 7.1 | 16.4 | 14.1 | 20.3 | 23.4 | 14.1 | 15.6 | 15.6 | 14.1 | 16.7 (32.0) |
| GRPO | 11.0 | 26.3 | 14.3 | 21.9 | 8.6 | 16.4 | 20.6 | 35.3 | 35.3 | 23.5 | 23.5 | 26.5 | 11.8 | 25.2 (17.0) |
| GSPO | 10.0 | 30.6 | 13.1 | 21.1 | 11.4 | 17.2 | 23.8 | 33.3 | 40.5 | 16.7 | 21.4 | 31.0 | 9.5 | 25.2 (21.0) |
| Reinforce++ | 11.2 | 28.8 | 14.5 | 20.3 | 9.2 | 16.8 | 21.6 | 31.5 | 33.5 | 20.5 | 21.6 | 22.8 | 12.7 | 23.5 (23.0) |
| **ArenaRL** | **32.1** | **66.1** | **31.7** | **58.0** | **21.0** | **41.8** | **62.6** | **77.3** | **78.8** | **57.1** | **55.6** | **57.1** | **61.6** | **64.3 (99.0)** |

Single-Elimination for all subsequent experiments.

## 6.3. Main Results

As summarized in Table 3, ArenaRL demonstrates strong performance and outperforms four closed-source models. When using Qwen3-8B-Base as backbone model, ArenaRL achieves a 41.8% average win rate on the Open-Travel benchmark, substantially surpassing GRPO (16.4%) and GSPO (17.2%). On Open-DeepResearch benchmark, it reaches a 64.3% win rate with valid generation rate (Val.,%) of 99%. In contrast, the SFT baseline struggles with a 32% Val.,%. We attribute this to the inherently high token consumption required by deepresearch tasks, together with the prevalence of long-horizon samples in the SFT training data. Notably, while GRPO and GSPO slightly improve win rates, their Val.,% fall below the SFT baseline. This indicates that pointwise reward schemes fail to capture fine-grained improvement signals in long-horizon tasks and are susceptible to spurious signals like length bias. Conversely, ArenaRL's comparison-based signal provides more discriminative gradients, steering the policy toward robust reasoning.

Table 4 further validates the generality of ArenaRL on standard open-ended writing tasks. Across three benchmarks, ArenaRL maintains a substantial lead in overall average score, outperforming GRPO by 7.4%, GSPO by 8.1%, and Reinforce++ by 7.7%. In addition, ArenaRL surpasses two strong closed-source models, GPT-4o and Claude-3.7-Sonnet, further highlighting its superiority. These consistent gains across diverse scenarios indicate that ArenaRL systematically enhances both reasoning and expressive capabilities, making it highly effective for a broad range of open-ended generation tasks beyond tool-augmented agents.

## 6.4. Further Analysis

**Impact of Group Size $N$.** We further investigate the scalability of ArenaRL by varying the group size $N \in \{2, 4, 8, 16\}$, as shown in Figure 4 (a). The results show a clear monotonic performance improvement as the group size increases. Notably, even at $N = 2$, ArenaRL achieves a 20.8% win rate, outperforming the SFT baseline (16.4%). This confirms that even the most basic pairwise comparison setting can provide effective optimization gradients. The

*Table 4.* Performance comparison on open-ended writing task across three public benchmarks.

| Method | WritingBench | | | | | | HelloBench | | LongBench | Mean |
|---|---|---|---|---|---|---|---|---|---|---|
| | WB-A | WB-B | WB-C | WB-D | WB-E | WB-F | Summ. | Heur. | Quality | |
| *Closed-source Models* | | | | | | | | | | |
| GPT-4o | 67.9 | 66.3 | 68.6 | 70.0 | 70.7 | 72.2 | 84.3 | 89.1 | 90.4 | 75.5 |
| Grok-4 | 80.3 | 78.7 | 79.8 | 81.5 | 81.2 | 80.9 | 85.6 | 94.7 | 96.5 | 84.4 |
| Gemini-2.5-pro | 80.9 | 80.4 | 82.5 | 84.3 | 83.5 | 82.6 | 82.4 | 93.8 | 98.7 | 85.5 |
| Claude-3.7-Sonnet | 68.4 | 66.5 | 68.7 | 70.3 | 71.4 | 71.5 | 74.7 | 95.8 | 98.3 | 76.2 |
| *Backbone Model: Qwen3-4B-Base* | | | | | | | | | | |
| SFT | 68.1 | 64.8 | 65.4 | 60.8 | 66.6 | 65.3 | 56.1 | 78.6 | 81.4 | 67.5 |
| GRPO | 69.7 | 68.2 | 66.1 | 64.2 | 67.5 | 65.9 | 62.0 | 78.2 | 83.0 | 69.4 |
| GSPO | 70.4 | 67.9 | 67.2 | 64.6 | 67.9 | 66.4 | 60.8 | 79.4 | 82.1 | 69.6 |
| Reinforce++ | 68.9 | 67.1 | 65.0 | 62.8 | 68.4 | 66.4 | 58.4 | 80.1 | 82.7 | 68.9 |
| **ArenaRL** | **76.2** | **76.0** | **74.4** | **68.7** | **76.1** | **74.4** | **76.7** | **82.7** | **87.0** | **76.9** |
| *Backbone Model: Qwen3-8B-Base* | | | | | | | | | | |
| SFT | 70.7 | 69.4 | 67.9 | 63.7 | 69.7 | 70.6 | 63.4 | 82.4 | 85.5 | 71.5 |
| GRPO | 71.6 | 71.2 | 68.7 | 66.8 | 72.6 | 70.3 | 64.9 | 83.8 | 87.0 | 73.0 |
| GSPO | 71.6 | 70.7 | 68.9 | 66.1 | 72.3 | 69.8 | 64.0 | 81.8 | 85.2 | 72.3 |
| Reinforce++ | 71.1 | 71.4 | 68.2 | 66.5 | 73.1 | 70.6 | 64.3 | 84.2 | 85.0 | 72.7 |
| **ArenaRL** | **78.1** | **77.7** | **77.6** | **75.0** | **79.4** | **77.2** | **73.8** | **91.3** | **93.8** | **80.4** |

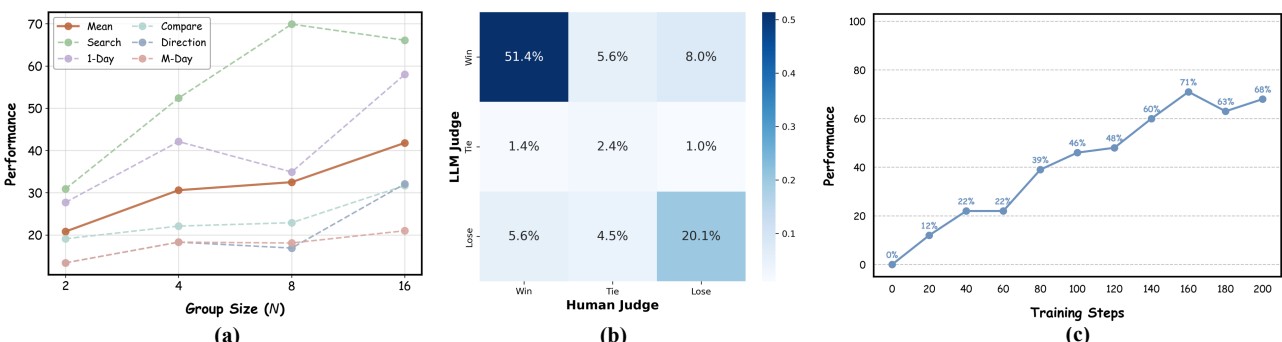

*Figure 4.* **(a)** The impact of Group Size $N$ on performance of Open-Travel benchmark. **(b)** The consistency between LLM and human evaluations. **(c)** The performance trend of ArenaRL in training Qwen3-8B via direct RL without cold start.

most significant gain occurs at $N = 16$, where the win rate reaches 41.8%. These results indicate that enlarging the group size effectively broadens the exploration space, increasing the likelihood of discovering high-quality trajectories, enabling the model to learn from stronger examples.

**Assessment of Consistency.** To assess the reliability of the LLM-based evaluation mechanism, we analyzed the consistency between LLM and human evaluations on the Open-Travel and Open-DeepResearch benchmarks using a confusion matrix (see Figure 4 (b)). The results show that most evaluation outcomes are concentrated along the diagonal, with an overall agreement rate of 73.9%. This relatively high level of consistency suggests that ArenaRL's performance gains do not simply stem from overfitting to the preferences of the specific judge model used during the RL phase, but instead reflect improvements that are broadly aligned with human assessments.

**Direct RL Training without Cold Start.** To further test

ArenaRL's generalizability, we bypassed the cold-start phase on the Open-Travel task and directly employed the Qwen3-8B model for ArenaRL training, reporting its performance on the *Search* sub-task. As shown in Figure 4 (c), the model's performance exhibited a stable upward trend. At the initial step 0, the model scored 0, indicating that the generic model was initially incapable of handling such complex travel planning tasks. However, as RL training steps increased, the model rapidly acquired the corresponding tool invocation capabilities. Ultimately, it achieves a peak score of 71% at step 160. This proves that ArenaRL's relative ranking mechanism can extract effective optimization signals even from low-quality initial outputs, enabling self-evolution from scratch without costly annotated data.

**Cross-Judge Robustness.** We further evaluate whether ArenaRL's gains are robust to the choice of reward model. As shown in Table 5, ArenaRL consistently outperforms the SFT baseline when using GPT-4.1, Claude-4-Sonnet,

*Table 5.* Cross-judge robustness of ArenaRL.

| Model | Open-Travel | | | | | Mean |
|---|---|---|---|---|---|---|
| | Direction | Search | Compare | 1-Day | M-Day | |
| SFT | 10.6 | 29.7 | 14.1 | 20.4 | 7.1 | 16.4 |
| GPT-4.1 | 27.4 | 57.3 | 32.5 | 53.8 | 18.4 | 37.9 |
| Claude-4-Sonnet | **35.2** | 64.3 | **34.6** | 57.4 | **24.6** | **43.2** |
| Qwen3-Max | 32.1 | **66.1** | 31.7 | **58.0** | 21.0 | 41.8 |

or Qwen3-Max as the judge for tournament-based ranking. The average win rate improves from 16.4% for SFT to 37.9%, 43.2%, and 41.8%, respectively. This demonstrates that ArenaRL's improvement is not tied to a specific judge family. In addition, the stronger reward models generally yield better performance, suggesting that higher-quality pairwise preferences can provide more reliable ranking signals for policy optimization.

## 7. Conclusion

In this paper, we introduced ArenaRL, a novel reinforcement learning framework that shifts the optimization paradigm for open-ended agents from pointwise scalar scoring to intra-group relative ranking. By implementing a process-aware pairwise evaluation mechanism and systematically analyzing five tournament topologies, we identified Seeded Single-Elimination as the optimal scheme for balancing advantage estimation accuracy with training efficiency. Extensive experiments indicate that this competition-driven paradigm provides robust advantage signals, motivating agents to exhibit superior reasoning and planning capabilities.

## Acknowledgment

This work was supported by the National Natural Science Foundation of China (NSFC) under Grant 62476260, the Fundamental Research Funds for the Central Universities under Grant WK2100000057. This work was supported by Alibaba Research Intern Program. We also acknowledge Ailin Ren and Ning Guo from Amap, Alibaba Group, for their valuable contributions to this work.

## Impact Statement

This work significantly advances reinforcement learning for open-ended LLM agents by addressing the critical challenge of reliable reward acquisition in subjective, diverse, and hard-to-verify tasks. By replacing pointwise scalar scoring with tournament-based relative ranking, ArenaRL mitigates discriminative collapse and provides stable advantage signals for optimizing long-horizon agent behavior. In addition, its process-aware pairwise evaluation encourages improvements in reasoning coherence, tool-use effectiveness, and final response quality, thereby enhancing the reliability of

agentic decision-making. These advances can support more capable AI systems for real-world applications such as personalized travel planning, autonomous deep research, and complex tool-augmented problem solving, laying a foundation for scalable and trustworthy open-ended agent learning.

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

# Appendix of ArenaRL

In this appendix, we provide the details omitted in the main text, offering additional analyses and discussions.

- Appendix A: Comprehensive experimental details and evaluation metrics.
- Appendix B: Introduction of the other three tournament topologies.
- Appendix C: Benchmark construction pipeline and tool annotation.
- Appendix D: Additional experiment results and discussions of ArenaRL.
- Appendix E: Summary of prompts.

## A. More Details about Experiments

### A.1. Implementation Details

All experiments adhere to a standard "cold-start $\rightarrow$ RL" pipeline, which is widely adopted to alleviate reward collapse in the early stages of reinforcement learning.

**Cold-start phase.** We adopt Qwen3-4B-Base (Yang et al., 2025) and Qwen3-8B-Base (Yang et al., 2025) as the backbone model. For open-ended agent tasks (Open-Travel and Open-DeepResearch), the backbone is first supervisedly fine-tuned on the corresponding SFT datasets in order to acquire basic tool-use and planning abilities. For open-ended writing tasks, we randomly sample 10k instances from the DeepWriting-20K dataset (Wang et al., 2025a) for supervised fine-tuning.

The cold-start training is conducted using the TRL framework with DeepSpeed ZeRO-3. The model is trained for 3 epochs on $32 \times$ NVIDIA H20 GPUs, with a learning rate of $2 \times 10^{-5}$ and a per-GPU batch size of 1.

**RL phase.** In the RL stage, we continue training on the corresponding RL splits for the open-ended agent tasks. For open-ended writing, we sample an additional 10k examples from DeepWriting-20K, explicitly excluding those used in the SFT phase, and use them for reinforcement learning.

We implement ArenaRL based on the Slime framework (Zhu et al., 2025). To concentrate optimization on reasoning quality, we mask out tokens corresponding to environment feedback from the loss computation. For Open-Travel and open-ended writing, we set the group size to $N = 16$ and the number of groups to $K = 8$. For Open-DeepResearch, we reduce these to $N = 8$ and $K = 4$ to improve training efficiency. Across all RL experiments, we employ the Adam optimizer with a learning rate of $1 \times 10^{-6}$.

During RL training, we use Qwen3-Max (Yang et al., 2025) as the arena judge, leveraging its stronger reasoning capabilities to produce high-fidelity pairwise comparison signals; the full judging prompt is provided in Appendix E. All RL experiments are conducted on $8 \times$ NVIDIA H20 GPUs.

### A.2. Evaluation Metrics

For the Open-Travel and Open-DeepResearch benchmarks, we invoke Qwen3-Max (Yang et al., 2025) and Claude-4-Sonnet (Anthropic, 2025) as evaluation models. The outputs of the tested agents are pairwise compared and scored against the baseline trajectory for each test sample, with corresponding win rates calculated.

For the Open-Travel tasks, we report the win rate on each of the five subtasks as well as the average win rate. The evaluation rubrics encompass multiple dimensions, such as the breadth and matching quality of the reasoning trajectory, alongside the feasibility and clarity of the answer.

For the Open-DeepResearch tasks, we report the valid generation rate (Val. %). And within the subset of valid generations, the win rates of the candidate model against the baseline under each of the seven evaluation rubrics, along with the final

averaged win rate. Specifically, these rubrics capture complementary aspects of open-ended research capability: **Framework** (*Frm.*), assessing the structural completeness and logical coherence of the initial research plan; **Tool Usage** (*Tool.*), evaluating the appropriateness and efficiency of tool invocations and their alignment with the research workflow; **Coverage** (*Cov.*), measuring whether the retrieved information sufficiently covers the user's requirements; **Relevance** (*Rel.*), assessing how well the response addresses all user queries and constraints; **Accuracy** (*Acc.*), evaluating the factual correctness and internal consistency of the content; **Depth** (*Dep.*), measuring the level of analytical depth and coherence of the reasoning process; and **Clarity** (*Cla.*), assessing the organization, readability, and practical usability of the final output. The complete task prompt for the judge models is shown in Figure 9.

For open-ended writing, we adopt three complementary benchmarks: WritingBench (Wu et al., 2025a), HelloBench (Que et al., 2024), and LongBench-write (Bai et al.). Considering the subjective nature of open-ended generation, and following established protocols (Wang et al., 2025a), we use Claude-3.7-Sonnet (Anthropic, 2023) to score generation quality on WritingBench and LongBench-write, and GPT-4o (Achiam et al., 2023) for HelloBench.

### A.3. Writing Benchmarks

To ensure the comprehensiveness and multifaceted nature of our evaluation, we adopt three complementary benchmarks: WritingBench (Wu et al., 2025a), HelloBench (Que et al., 2024), and LongBench-Write (Bai et al.).

1. **WritingBench** (Wu et al., 2025a) is designed to assess models' domain-specific writing proficiency and controllability across six professional and creative domains: A (Academic & Engineering), B (Finance & Business), C (Politics & Law), D (Literature & Arts), E (Education), and F (Advertising & Marketing). It focuses in particular on the ability to comply with complex, multidimensional constraints.

2. **HelloBench** (Que et al., 2024) evaluates model performance on a diverse set of "in-the-wild" tasks derived from real user queries. Our analysis concentrates on two key subsets: HelloBench-Summ. (text summarization), which evaluates the ability to summarize long-form texts; and HelloBench-Heur. (heuristic text generation), which measures creative reasoning and stylistic fidelity in long-form narrative continuation.

3. **LongBench-write** (Bai et al.) is designed to evaluate a model's ability to produce coherent ultra-long texts (*e.g.*, on the order of 10,000 words), enabling us to assess the fundamental capacity to maintain topical consistency and global coherence as output length scales.

## B. Other tournament topologies

In addition to the two tournament topologies described in Section 4, we investigate three alternative topologies to explore the trade-off between ranking fidelity and computational overhead. These variants provide a comprehensive landscape for understanding the impact of competitive structures on advantage estimation.

**Anchor-Based Ranking.** To minimize computational overhead to the theoretical lower bound, we consider an Anchor-Based Ranking scheme. For a given query $x$, we first generate a deterministic reference trajectory, denoted as the quality anchor $\tau_{anc}$, using greedy decoding (Temperature=0). The remaining $N-1$ trajectories $\tau_i, \ i = \{1, 2, ..., N-1\}$ in group $\mathcal{G}$ are generated via high-entropy sampling (*e.g.*, Temperature=0.8) to ensure diverse exploration. Each exploratory trajectory $\tau_i$ is individually compared with the anchor trajectory $\tau_{anc}$,

$$(s_i, s_{anc}^i) = \mathcal{J}(u, x, \tau_i, \tau_{anc}) + \mathcal{J}(u, x, \tau_{anc}, \tau_i), \quad \forall i \in \{1, \ldots, N-1\} \tag{12}$$

The relative rank is then derived from the scalar score set $\{s_1, \ldots, s_{N-1}, \bar{s}_{anc}\}$, where $\bar{s}_{anc} = \frac{1}{N-1} \sum s_{anc}^i$. While this topology achieves a minimal complexity of $O(N)$, it suffers from a significant resolution collapse. Since exploratory trajectories are never compared directly, the mechanism effectively measures the "distance from the deterministic baseline" but fails to resolve the fine-grained advantage signals between two high-quality samples that both outperform the anchor. This lack of intra-group resolution often leads to ambiguity in ranking among suboptimal solutions.

**Double-Elimination Tournament.** Drawing from competitive gaming theory, we implement a Double-Elimination Tournament to enhance the robustness of ranking against stochastic judge errors. This structure bifurcates the competition into a Winners' Bracket and a Losers' Bracket, ensuring that a trajectory is only assigned a lower rank after sustaining two defeats. Let $\mathcal{B}_W$ and $\mathcal{B}_L$ denote the dual brackets; a trajectory $\tau$ losing in $\mathcal{B}_W$ is relegated to $\mathcal{B}_L$ for a second-chance evaluation.

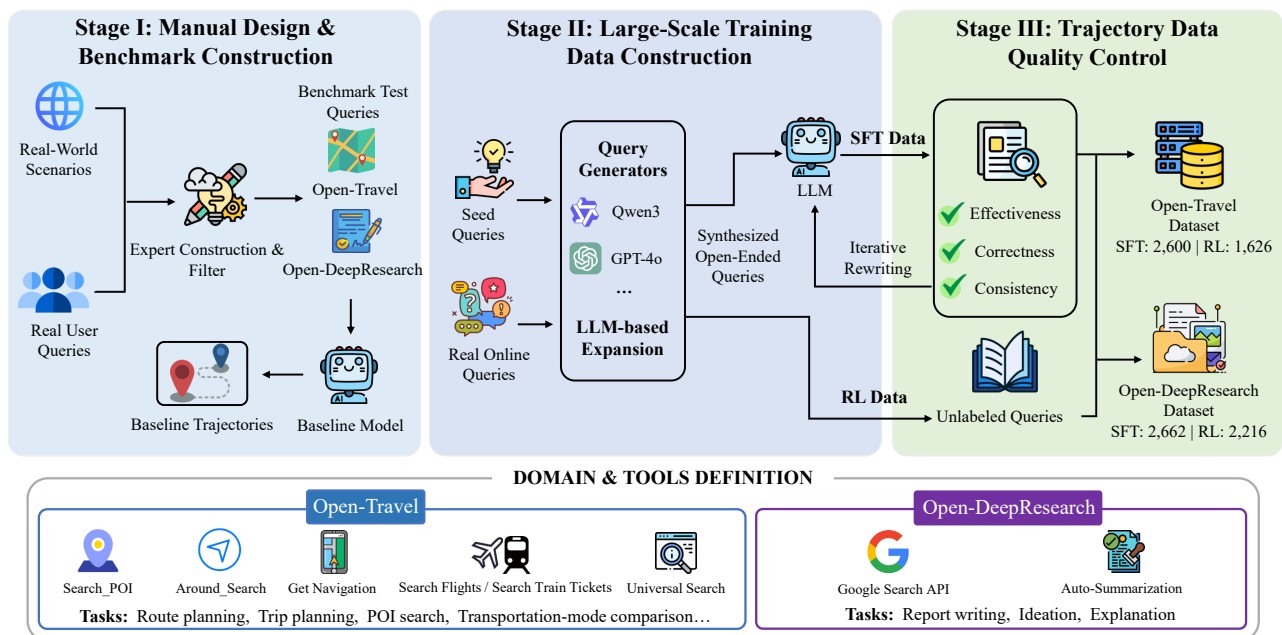

*Figure 5.* The construction pipeline of Open-Travel and Open-DeepResearch benchmarks.

The total comparison cost is approximately $2N - 2$, mirroring the budget of our Seeded Single-Elimination. However, without the benefit of an initial quality prior (anchor-based seeding), the Double-Elimination format relies on random initial pairings. Empirical results suggest that although this topology offers higher resilience against unexpected evaluations by the LLM judge, the absence of seeding frequently leads to premature collisions of top-tier trajectories. This leads to a marginal degradation in global ranking accuracy compared to Seeded Single-Elimination.

**Swiss-System.** The Swiss-System tournament is introduced as a non-eliminative alternative that maximizes the information extracted from a fixed number of rounds $K \approx \log_2 N$. In each round $k$, trajectories are matched using a balanced pairing protocol, where candidates with identical win-loss records are pitted against each other to maximize the discriminative utility of each match. The final ranking is determined by a composite metric $M(\tau_i)$:

$$M(\tau_i) = W(\tau_i) + \lambda \cdot \text{BH}(\tau_i), \quad \text{where } \text{BH}(\tau_i) = \sum_{\tau_j \in \mathcal{O}(\tau_i)} W(\tau_j) \tag{13}$$

Where $W(\tau)$ denotes the total wins and $\lambda$ is a tie-breaking coefficient. The Buchholz score $BH(\tau_i)$ signifies the sum of wins achieved against the set of opponents $\mathcal{O}(\tau_i)$ encountered by the trajectory $\tau_i$. This metric effectively accounts for the schedule strength, rewarding trajectories that performed well against strong opponents. Although the Swiss-System achieves $O(N \log N)$ complexity, its performance is still lower than the proposed seeded single-elimination scheme. We attribute this failure to the absence of an informative initial prior and insufficient comparison depth.

## C. More Details about Benchmarks

### C.1. Benchmark Construction Pipeline

Figure 5 further illustrates the data construction pipeline for the two benchmarks, Open-Travel and Open-DeepResearch. The pipeline comprises three main stages:

**Stage I: Benchmark Data Collection**    This phase establishes a rigorous evaluation foundation by curating high-quality, scenario-specific queries and generating reference trajectories to facilitate robust comparative analysis.

- **Open-ended query construction.** We begin by abstracting real-world application scenarios (such as travel planning and open-ended deep research) and collecting a corpus of authentic queries for each domain from publicly accessible

online platforms. These queries underwent rigorous anonymization by five domain experts, followed by multiple rounds of filtering and refinement to obtain a set of queries that are semantically precise and succinctly formulated. Finally, we select 50 queries for each of the five Open-Travel subtasks and 100 queries for Open-DeepResearch as the benchmark test sets.

- **Baseline trajectory construction.** To obtain a reference baseline for subsequent pairwise comparison and win-rate evaluation (*i.e.*, the basic agent capability level), we adopt Claude-Sonnet-4 (Anthropic, 2023) as the base model to generate complete tool-use trajectories and the corresponding open-ended answers.

**Stage II: Large-Scale Training Data Construction** After fixing the benchmark, we first manually construct a small set of representative seed queries tailored to the requirements of different sub-tasks in real application scenarios. We then use multiple LLMs with diverse styles (*e.g.*, Qwen3 (Yang et al., 2025), GPT-4o (Achiam et al., 2023)) as "query generators" to synthesize large-scale, multi-scenario open-ended queries. The resulting training dataset thus contains both authentic queries collected from public web platforms and diversified queries produced via LLM-based expansion.

Consistent with Stage I, we employ Qwen3-Max (Yang et al., 2025) and Claude-Sonnet-4 (Anthropic, 2023) as the base model to generate large-scale tool-use trajectories, which are used as cold-start data for SFT. The remaining queries, without explicit supervised trajectories, are reserved for subsequent RL.

**Stage III: Trajectory Data Quality Control** We first feed the complete trajectory dataset into a rule-augmented LLM-based quality inspection module. This module evaluates each trajectory along three dimensions: effectiveness of open-ended tool usage, correctness of conversational content, and consistency of the final answers. This process filters out trajectories with formatting or logical errors. For trajectories that fail to meet the criteria, we iteratively rewrite and refine them until they pass the LLM-based quality inspection.

### C.2. Benchmarking Tool Annotation

In this section, we will outline the tools employed by the Open-DeepResearch and Open-Travel benchmarks respectively.

**Open-DeepResearch.** We annotate commonly used search tools for open-ended deep research as candidate tools. Specifically, we perform web search via the Google API and integrate a summarization model into the search pipeline to prevent excessive context length. For each retrieved webpage, if the parsed content exceeds 2,500 characters, we invoke the Qwen3-Max (Yang et al., 2025) model to automatically summarize the page content.

**Open-Travel.** We annotate six commonly used tool types for travel planning:

- **Search poi:** This tool is built on Amap's POI search service and retrieves location information via text queries. The input can be a structured address (*e.g.*, "No. 10 Fuyong Street, Wangjing, Chaoyang District, Beijing") or a POI name (*e.g.*, "People's Square"). It returns multiple potentially relevant POIs, including: (1) detailed address; (2) geographic coordinates (the location field, formatted as "longitude, latitude"); and (3) business information (the Business field).

- **Around search:** This tool searches for POIs within a circular area specified by a center point and radius. Users can specify POI types or constrain the returned results using keywords (*e.g.*, "bank"). It returns multiple potentially relevant POIs, each including: (1) detailed address; (2) geographic coordinates (the location field, where longitude and latitude are separated by a comma, with longitude first and latitude second); and (3) business information.

- **Get navigation:** This tool provides integrated route planning based on Amap's navigation service, covering multiple transportation modes such as walking, driving, and public transit. The inputs are the latitude–longitude pairs of the origin and destination, with optional waypoints and a route-planning mode (driving by default). The tool returns a dictionary containing detailed route-planning information.

- **Universal search:** This tool performs general, city-level geographic knowledge search using the search engine. When users pose vague or open-ended questions (*e.g.*, "What are some fun things to do in Beijing?" or "What parks in Shanghai are suitable for family outings?"). It is used to obtain recommendations and travel suggestions. The tool returns the corresponding search results.

*Table 6.* Ablation study of different components of the seeded single-elimination on Open-Travel benchmark.

| Topology | Open-Travel | | | | | Mean |
|---|---|---|---|---|---|---|
| | Direction | Search | Compare | 1-Day | M-Day | |
| SFT | 10.6 | 29.7 | 14.1 | 20.4 | 7.1 | 16.4 |
| w/o Anchor | 30.4 | 58.5 | 28.6 | 53.2 | 18.7 | 37.9 |
| w/o Trajectory | 23.7 | 46.2 | 19.8 | 42.8 | 15.5 | 29.6 |
| w/o Bidirectional | 27.4 | 59.7 | 26.5 | 49.6 | 17.1 | 36.1 |
| w/o Seed | 28.6 | 51.0 | 23.7 | 45.1 | 18.2 | 33.3 |
| **ArenaRL** | **32.1** | **66.1** | **31.7** | **58.0** | **21.0** | **41.8** |

- **Search flights:** This tool searches for intercity flight information based on date and city names. When users need to query flights between two cities, they provide the query date and departure city, and the tool returns a list of flights, each including flight number, price, departure/arrival airports, and departure/arrival times.

- **Search train tickets:** This tool searches for intercity train ticket information based on date and city names. When users need to query train tickets between two cities, it returns a list of train options, where each entry includes train ID, price, departure/arrival stations, and times, and explicitly distinguishes between direct and transfer routes.

For these six tools, search poi, around search, and get navigation are implemented using Web service APIs from the Amap open platform[2]. The universal search tool is powered by the Bailian search engine[3]. The outputs of search flights and search train tickets are simulated by GPT-4.1-mini (OpenAI, 2025), with the detailed prompts provided in the Appendix E.

## D. More Results

### D.1. Summary of Experimental Results

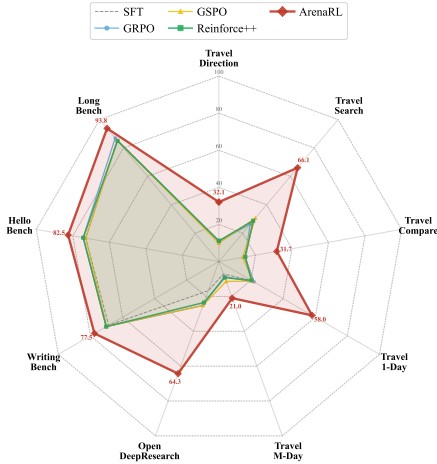

*Figure 6.* Performance overview of ArenaRL algorithm.

Figure 6 presents the comparison of ArenaRL and baseline methods on three categories of open-ended tasks (travel planning, deep research, and open-ended writing), when employing Qwen3-8B-Base as the backbone. The experimental results demonstrate that ArenaRL significantly outperforms SFT and existing RL algorithms across all evaluation dimensions. In particular, in the Open-Travel and Open-DeepResearch benchmarks, ArenaRL achieves nearly a threefold performance improvement over traditional RL algorithms such as GRPO. Furthermore, ArenaRL maintains its lead across three open-ended writing benchmarks. This further validates that ArenaRL is not only applicable to tool-augmented agents but also systematically enhances models' logical reasoning and long-text generation capabilities, exhibiting strong generalizability. These results indicate that, by shifting from pointwise scalar rewards to intra-group relative ranking, ArenaRL effectively mitigates the issue of reward signals being overwhelmed by noise in open-ended tasks, thereby providing a scalable RL paradigm for building high-performance open-ended agents.

### D.2. Ablation Study

To assess the contribution of each key component in ArenaRL, we conduct an ablation study under the seeded single-elimination tournament topology, with results in Table 6. Specifically, *w/o Anchor* denotes a variant that does not use greedy decoding as a quality anchor, instead selecting reference trajectories at random; *w/o Trajectory* evaluates only the final answers in pairwise comparisons, without assessing the intermediate reasoning trajectories; *w/o Bidirectional* disables the bidirectional evaluation mechanism; and *w/o Seed* removes seed-based pre-ranking, resorting directly to random matching.

---

[2]https://lbs.amap.com/api/webservice/guide/api-advanced/search
[3]https://bailian.console.aliyun.com/?spm=5176.12818093_47.resourceCenter.1.3dd916d04Ye5xn&tab=app#/mcp-market/detail/WebSearch

Among all variants, *w/o Trajectory* exhibits the most pronounced performance drop, with the average win rate decreasing from 41.8% to 29.6%. This indicates that, for long-horizon open-ended agentic tasks, relying solely on final answers tends to induce overfitting to superficial features, whereas incorporating process-aware constraints is crucial for maintaining efficient reasoning. The win rates of *w/o Seed* and *w/o Anchor* decline to 33.3% and 37.9%, respectively. This suggests that using the greedy-decoded trajectory as a quality anchor to construct a robust prior, together with designing the initial tournament bracket via seed-based pre-ranking, substantially improves the accuracy of ranking estimation. The performance of *w/o Bidirectional* is also inferior to the full ArenaRL system, reflecting that LLM-based evaluators typically exhibit positional bias in pairwise comparison, and that a bidirectional scoring protocol is essential for obtaining unbiased and reliable reward signals. In summary, the superiority of ArenaRL arises from the synergy among its components: it preserves linear computational complexity $O(N)$ while maximizing the quality and stability of relative ranking signals.

### D.3. Tournament Algorithm Flow

In Algorithm 1, we present the detailed procedure of the proposed seeded single-elimination tournament, which achieves an excellent trade-off between training efficiency and the accuracy of advantage estimation.

### D.4. Case Study

Figures E.6 and E.7 illustrate the interaction trajectories of the baseline SFT model and the model tuned via ArenaRL within a representative case under the Open-Travel benchmark. This example involves a complex travel request with multiple hard constraints, including time, destination, and budget. The baseline SFT model exhibits a restatement tendency in its chain-of-thought, and its reasoning trajectory fails to align with the user's intent, often overlooking specific constraints and providing only generic suggestions. In contrast, the model optimized with ArenaRL demonstrates strong strategic planning capabilities. It proactively retrieves information about multiple target attractions, performs logically coherent route planning, and ultimately produces a persuasive, personalized itinerary. This substantial improvement in reasoning patterns powerfully demonstrates that our tournament-based ranking mechanism effectively incentivizes the model to explore and retain superior planning strategies.

## E. Prompts

In this section, we will illustrate all the prompts used in our paper, with the Chinese prompts translated into English.

### E.1. Open-Travel Task Prompt

The pairwise evaluation prompt for ArenaRL is shown in Figure 7. Additionally, we illustrate the pointwise evaluation prompts for algorithms such as GRPO and GSPO using the Open-Travel task as an example, as depicted in Figure 8.

### E.2. Open-DeepResearch Task Prompt

See Figure 9.

### E.3. Open-ended Writing Task Prompt

See Figure 10.

### E.4. Search Flights Tool Prompt

See Figure 11.

### E.5. Search Train Tickets Tool Prompt

See Figure 12.

---

**Algorithm 1** ArenaRL: Seeded Single-Elimination Advantage Estimation

---

**Input:** Trajectory group $\mathcal{G} = \{\tau_{anc}\} \cup \{\tau_1, \ldots, \tau_{N-1}\}$, Arena Judge $\mathcal{J}(\cdot, \cdot)$
**Output:** Standardized advantages $\mathcal{A} = \{A_1, \ldots, A_N\}$
   `// Phase 1: Anchor-Based Seeding`
1   Initialize score set $\mathcal{S}_{init} \leftarrow \emptyset$
2   **for** $i \leftarrow 1$ **to** $N-1$ **do**
3      Obtain scores: $s_i, s_{anc}^i \leftarrow \mathcal{J}(\tau_i, \tau_{anc})$
4      $\mathcal{S}_{init} \leftarrow \mathcal{S}_{init} \cup \{(\tau_i, s_i)\}$
5   **end**
6   $s_{anc} \leftarrow \frac{1}{N-1} \sum_{i=1}^{N-1} s_{anc}^i; \quad \mathcal{S}_{init} \leftarrow \mathcal{S}_{init} \cup \{(\tau_{anc}, s_{anc})\}$
7   Sort $\mathcal{G}$ descendingly based on $\mathcal{S}_{init}$ to get ordered seeds: $\mathcal{P} \leftarrow [\tau_1, \ldots, \tau_N]$
8   Initialize accumulated scores $V[\tau] \leftarrow$ score in $\mathcal{S}_{init}$ for all $\tau \in \mathcal{G}$
9   Initialize ranking tiers $\mathcal{T} \leftarrow \emptyset$
   `// Phase 2: Elimination Tournament`
10   Initialize array $\mathcal{B}$ of size $N$
11   $idx_{head} \leftarrow 1; \quad idx_{tail} \leftarrow N-1$                          `// Tail pointer at start of last pair`
12   **for** $k \leftarrow 1$ **to** $N/2$ **do**
13      $(\tau_{high}, \tau_{low}) \leftarrow (\mathcal{P}[k], \mathcal{P}[N-k+1])$                  `// Pair Seed` $k$ `vs` $N-k+1$
14      **if** $k$ *is odd* **then**
15          $\mathcal{B}[idx_{head}] \leftarrow \tau_{high}; \quad \mathcal{B}[idx_{head}+1] \leftarrow \tau_{low}$
16          $idx_{head} += 2$                               `// Fill from front`
17      **else**
18          $\mathcal{B}[idx_{tail}] \leftarrow \tau_{high}; \quad \mathcal{B}[idx_{tail}+1] \leftarrow \tau_{low}$
19          $idx_{tail} -= 2$                                `// Fill from back`
20      **end**
21   **end**
22   **while** $|\mathcal{B}| > 1$ **do**
23      $\mathcal{W}_{round} \leftarrow \emptyset; \quad \mathcal{L}_{round} \leftarrow \emptyset$
24      **for** $k \leftarrow 1$ **to** $|\mathcal{B}|/2$ **do**
25          $(\tau_a, \tau_b) \leftarrow (\mathcal{B}[2k-1], \mathcal{B}[2k]); \quad (s_a, s_b) \leftarrow \mathcal{J}(\tau_a, \tau_b)$
26          $V[\tau_a] += s_a; \quad V[\tau_b] += s_b$
27          $(\tau_{win}, \tau_{lose}) \leftarrow (s_a > s_b)?(\tau_a, \tau_b):(\tau_b, \tau_a)$               `// Determine result`
28          $\mathcal{W}_{round} \leftarrow \mathcal{W}_{round} \cup \{\tau_{win}\}; \quad \mathcal{L}_{round} \leftarrow \mathcal{L}_{round} \cup \{\tau_{lose}\}$
29      **end**
30      $\mathcal{T} \leftarrow \mathcal{T} \cup \{\mathcal{L}_{round}\}; \quad \mathcal{B} \leftarrow \mathcal{W}_{round}$           `// Archive losers, promote winners`
31   **end**
32   $\mathcal{T} \leftarrow \mathcal{T} \cup \{\mathcal{B}\}$                                   `// Champion enters final tier`
   `// Phase 3: Rank Assignment (Hierarchical Sorting)`
33   $k \leftarrow 0$
34   **foreach** $Tier \in \mathcal{T}.reverse()$ **do**
35      Sort $Tier$ descendingly based on accumulated scores $V$
36      Assign Rank$(\tau_i) \leftarrow k++$ for all $\tau_i \in Tier$
37   **end**
   `// Phase 4: Advantage Calculation`
38   Assign Reward: $r_{\tau_i} \leftarrow 1 - \frac{\text{Rank}(\tau_i)}{N-1}$
39   Compute statistics $\mu_r, \sigma_r$ from $\{r_{\tau_i}\}$
40   **return** $A_i \leftarrow \frac{r_{\tau_i} - \mu_r}{\sigma_r + \epsilon}$ for all $\tau_i \in \mathcal{G}$

---

## E.6. Case Study of SFT model

---

**Case Trajectory in Open-Travel Benchmark of SFT model (translated into English)**

**Query:** A two-day weekend hiking trip from Chengdu to Qingcheng Back Mountain for two people, with a focus on natural scenery and photography, and a budget of 600 RMB per person.

**Step 1**

**<think>**
The user would like to plan a weekend hiking trip from Chengdu to Qingcheng Back Mountain for two people, with a preference for natural scenery and photography, and a budget of 600 RMB per person. I need to first determine the location of Qingcheng Back Mountain, then plan transportation from Chengdu to that area, as well as hiking routes in Qingcheng Back Mountain.
First, I need to use the search_poi tool to look up the location information of Qingcheng Back Mountain.
**</think>**
**<tool_call>**
{"name": "search_poi", "arguments": {"query": "Qingcheng Back Mountain"}}
**</tool_call>**
**<tool_response>**
...
**</tool_response>**

**Step 2**

**<think>**
I need to plan a weekend hiking itinerary from Chengdu to Qingcheng Back Mountain for the user, suitable for two people, with a preference for natural scenery and photography, and a budget of 600 RMB per person. I have already obtained the basic location information for Qingcheng Back Mountain, and now need to further understand the transportation options from Chengdu to Qingcheng Back Mountain, as well as the hiking routes in Qingcheng Back Mountain and the surrounding natural scenery and photography spots.
First, I need to find out the transportation options from Chengdu to Qingcheng Back Mountain.
**</think>**
**<tool_call>**
{"name": "universal_search", "arguments": {"query": "Transportation options from Chengdu to the Qingcheng Back Mountain"}}
**</tool_call>**
**<tool_response>**
...
**</tool_response>**

**Step 3**

**<think>**
I need to plan a weekend hiking itinerary from Chengdu to Qingcheng Back Mountain for the user, suitable for two people, focusing on natural scenery and photography, with a budget of 600 RMB per person. I have already obtained basic information and transportation options for Qingcheng Back Mountain, and now need to further understand the hiking routes and points of interest in Qingcheng Back Mountain in order to design a detailed schedule.
**</think>**
**<tool_call>**
{"name": "universal_search", "arguments": {"query": "Hiking Routes and Prime Photography Spots on Qingcheng Back Mountain"}}
**</tool_call>**
**<tool_response>**
...
**</tool_response>**

**Step 4**

**<think>**
I need to plan a weekend hiking itinerary from Chengdu to Qingcheng Back Mountain for the user, with an emphasis

---

on natural scenery and photography, and a budget of 600 RMB per person. I have already obtained the location information and some basic data for both Chengdu and Qingcheng Back Mountain, and now need to further understand the location of Chengdu in order to plan the transportation route.
**</think>**
**<tool_call>**
{"name": "search_poi", "arguments": {"query": "Chengdu"}}
**</tool_call>**
**<tool_response>**
...
**</tool_response>**

**Step 5**

**<think>**
I need to plan a weekend hiking itinerary from Chengdu to Qingcheng Back Mountain for the user, with a focus on natural scenery and photography, and a budget of 600 RMB per person. I have already obtained the location information of Chengdu and Qingcheng Back Mountain, as well as basic information about transportation and hiking routes. Now I need to design a specific travel plan and related details.
First, I need to determine the specific transportation route from Chengdu to Qingcheng Back Mountain. Based on previous search results, there are several main options for traveling from Chengdu to Qingcheng Back Mountain:
1. Take a high-speed train from Chengdu to Qingchengshan Station, then transfer to a scenic area shuttle bus or taxi to get to the Back Mountain area;
2. Drive directly to Qingcheng Back Mountain.
I will use the get_navigation tool to plan the transportation route from Chengdu to Qingcheng Back Mountain.
**</think>**
**<tool_call>**
{"name": "get_navigation", "arguments": {"start_lon": 104.066301, "start_lat": 30.572961, "end_lon": 103.487136, "end_lat": 30.93071}}
**</tool_call>**
**<tool_response>**
...
**</tool_response>**

**Step 6**

**<answer>**
Chengdu to Qingcheng Rear Mountain Hiking Guide (Nature Landscape Photography Trip for Two)
Trip Overview Total budget: 600 RMB per person (1,200 RMB for two) Duration: 2 days over a weekend (depart Saturday, return Sunday) Group size: 2 people Main activities: Hiking, nature landscape photography, leisure sightseeing
Detailed Itinerary Day 1: Chengdu → Qingcheng Rear Mountain Morning: Depart from Chengdu 07:00–08:00 Depart from Chengdu for Qingcheng Mountain
Recommended transport: High-speed train from Chengdu Xipu Station to Qingchengshan Station Duration: about 30 min Fare: 10 RMB Alternative: Self-drive Duration: about 1.5 hours Distance: 50 km 08:30–09:00 After arriving at Qingchengshan Station, head to the Tai'an Ancient Town parking lot
Take the scenic-area shuttle bus (16 RMB/person) or taxi (50 RMB/person) to the Rear Mountain entrance Buy Rear Mountain entrance tickets (20 RMB/person) Midday: Lunch in the Scenic Area 09:30–11:00 Lunch inside the scenic area Recommended: Eat at Youyicun (Youyi Village) or Tai'an Ancient Town Try local specialties such as smoked pork ribs and potatoes Budget: About 30–50 RMB per person Afternoon: Qingcheng Rear Mountain Hike 11:00–18:00 Hiking route on Qingcheng Rear Mountain
Route: Wulong Valley → Youyi Village → Baiyun Cave → Cuiying Lake → Feiquan Valley → Tai'an Ancient Town Route difficulty: Recommended Route 2 (about 70% difficulty), includes boat ride, cable car, and hiking Main sights: Wulong Valley: Beautiful scenery, best starting point for the hike Youyi Village: Small mountain village with rest stops and snacks Baiyun Cave: Worth exploring Cuiying Lake: Boat ride available (3 RMB/person), very scenic Feiquan Valley: Flowing streams and natural landscapes Cable car option: Baiyun Cableway (45 RMB/person one way) to save energy Photography tips: The section from Wulong Valley to Youyi Village has the best views Cuiying

Lake and Feiquan Valley are also great for photography 18:00–19:00 Return to the scenic area entrance

Take a sightseeing bus (10 RMB/person) to the visitor center, then shuttle bus to Qingchengshan Station Or take a taxi ( 20 RMB/person) directly to Qingchengshan Station Evening: Return to Chengdu 19:30–21:00 Take the high-speed train back to Chengdu Day 2: In-Depth Exploration & Photography Morning: Depart from Chengdu 07:00–08:00 Depart from Chengdu for Qingcheng Mountain Same transport options as Day 1 Midday: Lunch in the Scenic Area 08:30–09:30 Lunch in the scenic area Same recommendations as Day 1 Afternoon: In-Depth Hiking on Qingcheng Rear Mountain 09:30–17:00 Deeper exploration of Qingcheng Rear Mountain

Route choice: You can try Route 3 (100% difficulty). Use caution; suitable for hikers with some experience. Key sights: Wanfo Cave: A whole rock wall covered with Buddha statues, very impressive Qianfo Cave: Also worth visiting Santan Wuquan (Three Pools and Misty Springs): A must-see classic spot Photography tips: Afternoon light is better for shooting caves and pools 17:00–18:00 Return to the scenic area entrance

Same transport options as Day 1 Evening: Return to Chengdu 18:30–20:00 Take the high-speed train back to Chengdu Cost Breakdown (Total 1,200 RMB for Two) Transportation High-speed train: Chengdu–Qingchengshan round trip About 40 RMB/person In-park transport: Scenic-area shuttle bus/taxi About 30–50 RMB/person Return transport: From Qingchengshan back to Chengdu About 40–60 RMB/person Tickets Qingcheng Rear Mountain entrance: 20 RMB/person (40 RMB for two) Cable car: Baiyun Cableway 45 RMB/person one way (round trip for two: 180 RMB, if needed) Cuiying Lake boat: 3 RMB/person (6 RMB for two) Food Meals in the scenic area: About 30–50 RMB/person (60–100 RMB for two) Snacks & supplies: Water, energy snacks, etc. About 50 RMB/person (100 RMB for two) Accommodation (If Extending the Trip) In-park lodging: Zhuolan Villa or local guesthouses About 200–300 RMB/night Recommended: Lanxipan Guesthouse in Dujiangyan Offers pick-up/drop-off and has a pleasant environment Photography Gear Camera: 800–1,000 RMB for two if renting (about 200–300 RMB/person) Lenses: About 500–800 RMB for two (if renting) Other gear: Tripod, filters, memory cards, etc. About 200–300 RMB/person Practical Tips Best visiting time

June–September: 08:00–18:00 (Mon–Sun) January–May & October–December: 08:00–17:30 Hiking gear

Comfortable, non-slip hiking shoes (essential) Lightweight backpack Drinking water and energy snacks Sun protection (hat, sunscreen, etc.) Camera tripod (essential for photography) Photography suggestions

Early morning and late afternoon light is best for landscapes After rain, Qingcheng Mountain is often shrouded in mist—excellent for photos Cuiying Lake and Feiquan Valley are popular photography spots Notes & safety

Mountain trails are steep; be careful and consider bringing trekking poles Toilets are scarce on the mountain; use water sparingly Mountain temperatures are lower than in the city; bring a jacket Check the weather forecast in advance and avoid heavy rain Keep your phone charged and communication available Wish you a wonderful trip and beautiful photos of Qingcheng Mountain!

**</answer>**

## E.7. Case Study Following ArenaRL

---

**Case Trajectory in Open-Travel Benchmark Following ArenaRL (translated into English)**

**Query:** A two-day weekend hiking trip from Chengdu to Qingcheng Back Mountain for two people, with a focus on natural scenery and photography, and a budget of 600 RMB per person.



**Step 1**



**<think>**
The user wants to learn about a hiking itinerary from Chengdu to Qingcheng Back Mountain, suitable for two people, with a preference for natural scenery and photography, and a budget of 600 RMB per person. I need to first obtain basic information about Qingcheng Back Mountain, and then plan the travel route.
**</think>**
**<tool_call>**
{"name": "universal_search", "arguments": {"query": "Hiking Route from Chengdu to the Qingcheng Back Mountain, Natural Scenery, Photography"}}
**</tool_call>**
**<tool_response>**
...
**</tool_response>**



**Step 2**



**<think>**
I have already obtained the basic information and transportation options for hiking from Chengdu to Qingcheng Back Mountain. Now I need to further look up more detailed information, including the exact location of Qingcheng Back Mountain and the locations of its main scenic spots, in order to plan a reasonable hiking route. First, I need to use the search_poi tool to find the location information of Qingcheng Back Mountain.
**</think>**
**<tool_call>**
{"name": "search_poi", "arguments": {"query": "Qingcheng Back Mountain Scenic Area"}}
**</tool_call>**
**<tool_response>**
...
**</tool_response>**



**Step 3**



**<think>**
I have already obtained the basic information and location of Qingcheng Back Mountain. Now I need to continue searching for more information about the locations of the main scenic spots inside Qingcheng Back Mountain, in order to plan a reasonable hiking route. According to previous search results, the main attractions of Qingcheng Back Mountain include Tai'an Ancient Town, Wulong Valley, Youyicun, Cuiying Lake, etc. I need to look up the exact location of each of these spots one by one.
**</think>**
**<tool_call>**
{"name": "search_poi", "arguments": {"query": "Tai'an Ancient Town at the Qingcheng Back Mountain"}}
**</tool_call>**
**<tool_response>**
...
**</tool_response>**



**Step 4**



**<think>**
I have already obtained the basic information of Qingcheng Back Mountain and the location of Tai'an Ancient Town. Now I need to continue searching for the location of Wulong Valley, because it is one of the most important scenic spots in Qingcheng Back Mountain.
**</think>**
**<tool_call>**

---

{"name": "search_poi", "arguments": {"query": "Wulong Valley at the Qingcheng Back Mountain"}}
**</tool_call>**
**<tool_response>**

...
**</tool_response>**

**Step 5**

**<think>**
I have already obtained the basic information of Qingcheng Back Mountain, as well as the location information of Tai'an Ancient Town and Wulong Valley. Now I need to continue searching for other main attractions, such as Youyicun, Cuiying Lake, etc., in order to plan a complete hiking route.
**</think>**
**<tool_call>**
{"name": "search_poi", "arguments": {"query": "Youyicun at the Qingcheng Back Mountain"}}
**</tool_call>**
**<tool_response>**

...
**</tool_response>**

**Step 6**

**<think>**
I have obtained the location information of the main attractions of Qingcheng Back Mountain, including Tai'an Ancient Town, Wulong Valley, and Youyicun. Now I need to continue looking up the location of Cuiying Lake, and then plan a complete hiking route.
**</think>**
**<tool_call>**
{"name": "search_poi", "arguments": {"query": "Cuiying Lake at the Qingcheng Back Mountain"}}
**</tool_call>**
**<tool_response>**

...
**</tool_response>**

**Step 7**

**<think>**
I have collected the location information of the main attractions of Qingcheng Back Mountain, including Tai'an Ancient Town, Wulong Valley, Youyicun, and Cuiying Lake. Now I need to continue searching for some other important attractions, such as Laifeng Pavilion and Water Curtain Cave, in order to plan a complete hiking route.
**</think>**
**<tool_call>**
{"name": "search_poi", "arguments": {"query": "Laifeng Pavilion at the Qingcheng Back Mountain"}}
**</tool_call>**
**<tool_response>**

...
**</tool_response>**

**Step 8**

**<think>**
I have collected the location information of the main scenic spots in Qingcheng Back Mountain, including Tai'an Ancient Town, Wulong Valley, Youyicun, Laifeng Pavilion, and Cuiying Lake. Now I need to plan a reasonable hiking route and provide detailed navigation information. Based on the information collected, I will plan a hiking route starting from Tai'an Ancient Town, passing through Wulong Valley, Youyicun, Laifeng Pavilion, and Cuiying Lake, and finally returning to Tai'an Ancient Town. In this way, the route can cover the main natural scenic spots of Qingcheng Back Mountain and is suitable for photography enthusiasts.
Next, I need to use the get_navigation tool to obtain the navigation information for this route.
**</think>**
**<tool_call>**

{"name": "get_navigation", "arguments": {"start_lon": 103.490619, "start_lat": 30.920645, "end_lon": 103.490619, "end_lat": 30.920645, "via_points": "103.473101,30.923365;103.473823,30.934175;103.476093,30.931739;103.483087,30.936879"}}
**</tool_call>**
**<tool_response>**
...
**</tool_response>**

| Step 9 |
|---|

**<answer>**
Chengdu to Qingcheng Rear Mountain Hiking Guide (Weekend 2-Day Trip)

Trip Overview Qingcheng Rear Mountain is one of the best hiking destinations around Chengdu, famous for its natural scenery and tranquil environment, and is especially suitable for photography enthusiasts. Below is a detailed 2-day itinerary to help you fully experience the beauty of Qingcheng Rear Mountain.

Transportation Getting to Qingcheng Mountain High-speed train: Take a high-speed train from Chengdu Xipu Station to Qingchengshan Station (about 30 minutes, 10 RMB per person one way). Self-drive: About 2 hours from Chengdu. Navigate to "Qingchengshan Rear Mountain Scenic Area Parking Lot" (there are Parking Area A and Parking Area B). Transport within the Scenic Area After getting off at Qingchengshan Station, you'll find tourist shuttle buses to the Front and Rear Mountains on the left side outside the station. Rear Mountain costs 15 RMB per person one way (about 40 minutes). Inside the scenic area there are sightseeing shuttle buses (10 RMB per person), or you can walk. Tickets Adult entrance ticket: 20 RMB per person Student ticket: 10 RMB per person Sightseeing shuttle bus in the scenic area: 10 RMB per person (one way) Luggage storage: 5 RMB per use Accommodation Suggestions Inside the Scenic Area Zhuolan Shanzhuang (Zhuolan Mountain Villa): Located at the foot of the mountain, offers mountain views, good service, and pick-up/drop-off service. Qixin Shanyin Guesthouse: Located in Taian Ancient Town, with a beautiful environment and attentive service. Youyicun Yuchengyuan: Located inside the Youyicun scenic area, quiet and secluded surroundings. Outside the Scenic Area There are many guesthouses and farmhouse inns in Taian Ancient Town, generally 200–500 RMB per night. Booking in advance is recommended. Food and Dining Dining in the Scenic Area There are restaurants and snack stalls in the scenic area, but prices are relatively high. It's recommended to eat in the ancient town, where you can find 2-person set meals for around 30 RMB (one meat dish, one vegetable dish, one soup). Local specialties: farmhouse dishes, wild mushrooms from Qingcheng Mountain, free-range chicken, corn, etc. Bringing Your Own Food There are water sources and small shops along the hiking route, so you can bring some snacks and water. It's not recommended to carry too much food—keep your pack as light as possible. Recommended Hiking Route Route Overview Taian Ancient Town → Wulong Valley → Youyicun (Another Village) → Laifeng Pavilion → Cuiying Lake → Taian Ancient Town

Detailed Itinerary Day 1

Morning

Depart from Chengdu for Qingcheng Mountain (by high-speed train or by car). After arriving at Qingchengshan Station, take the tourist shuttle bus to the Rear Mountain scenic area (about 40 minutes). Buy tickets at the scenic area entrance (20 RMB per person). Start hiking from Taian Ancient Town, following Qingcheng Mountain Road.

Midday

Arrive at Wulong Valley, which features waterfalls, mountain springs, and ancient plank roads with beautiful scenery. Have lunch and rest near Wulong Valley (you can bring your own food or eat at a restaurant in the scenic area).

Afternoon

Continue hiking to Youyicun; the scenery along this stretch is very pleasant. Visit the Youyicun scenic spot and enjoy the mountain and water views. Head to Laifeng Pavilion, an excellent viewpoint for sightseeing and photography. On the way, enjoy the scenery of Cuiying Lake; the water is clear and the environment serene. Return to Taian Ancient Town in the evening and have dinner there. Day 2

Morning

Depart from Taian Ancient Town and hike along Qingcheng Mountain Road. Focus on exploring Wulong Valley, the most scenic section of the Rear Mountain, with many waterfalls and mountain springs. You can choose to take a bamboo raft in Wulong Valley for a different way of sightseeing. Midday

Have lunch and rest in Youyicun, and sample local farmhouse specialties. Afternoon

Continue hiking to Cuiying Lake; the water is crystal clear and the surroundings are tranquil. Stay at Cuiying Lake for photos and to enjoy the reflections of the mountains in the water. Return to Taian Ancient Town in the evening, ending the trip. Recommended Photography Spots Wulong Valley: The most beautiful section of the Rear Mountain, with waterfalls, springs, and ancient plank roads. Bamboo rafting is available. The scenery is serene and highly photogenic. Laifeng Pavilion: Excellent vantage point for panoramic shots of the surrounding mountains and waters. Cuiying Lake: Clear water and tranquil surroundings, ideal for shooting reflections. Youyicun: A rustic village integrated with natural scenery, combining cultural and natural elements. Suggested Hiking Gear Comfortable hiking shoes—mountain trails are rugged, so footwear comfort is very important. Bring trekking poles—some sections are steep and narrow; the left-hand ascent offers better photo spots and can be tricky. Pack light; avoid unnecessary items. Bring enough drinking water, but note that there are very few toilets in the scenic area, so don't drink excessively. Bring a camera and spare batteries—Qingcheng Rear Mountain's natural scenery is excellent for photography. Practical Tips Best time to start: Set out around 8:30–9:00 a.m. to avoid peak crowds. Descent timing: Keep an eye on time; going down usually takes longer than going up. Weather: Check the forecast; trails can be slippery in the rain, so prepare for traction and safety. Sun protection: UV can be strong in the mountains; use sunscreen and wear a hat. Warm clothing: Even in summer it can be cool on the mountain; bring a light jacket. Pack light: Minimize what you carry so you can move easily. Environmental awareness: Take your trash with you and protect the natural environment. Budget Reference (Per Person) Transportation: Round-trip high-speed train 20 RMB + tourist shuttle bus 30 RMB = 50 RMB Tickets: Scenic area ticket 20 RMB + sightseeing shuttle bus 10 RMB (round trip) = 40 RMB Accommodation: 200–500 RMB (depending on guesthouse or farmhouse inn) Food: 30 RMB/day (for two people) × 2 days = 120 RMB Gear: Hiking shoes, trekking poles, photography gear, etc. about 100 RMB Other: Water, snacks, small souvenirs, etc. about 50 RMB Total: Approximately 460–610 RMB per person

This budget is for reference only; actual costs will vary based on accommodation choices and dining preferences. If you stay outside the scenic area or purchase equipment in advance, your total expenses may be lower.

Wish you a pleasant trip and wonderful photos of Qingcheng Rear Mountain!

**</answer>**

You are a "Travel Planning LLM Agent Comprehensive Reviewer," deeply versed in the travel industry and possessed of rigorous logic and evaluation methodology. Your task is to perform a multi-dimensional quantitative evaluation of the reasoning paths (Path) and response results (Answer) of LLM Agent A and Agent B under the same User Query, providing final composite scores and determining a winner. Please strictly follow the indicators, scoring rules, and output format below.

**I. Evaluation Content Format**
<USER_QUERY>{User's original question}</USER_QUERY>

<PATH_A>{Full reasoning path of LLM Agent A}</PATH_A>

<PATH_B>{Full reasoning path of LLM Agent B}</PATH_B>

<ANSWER_A>{Full response of LLM Agent A}</ANSWER_A>

<ANSWER_B>{Full response of LLM Agent B}</ANSWER_B>

**II. Path Evaluation**
【Dimension Descriptions】
1. Breadth: Whether it comprehensively covers user needs while avoiding redundant or repetitive steps.
2. Relevance: The degree to which each step aligns with the user's core requirements.
3. Detail: Whether the cited facts, data, timestamps, costs, reservation rules, and other details are sufficient, accurate, and useful.

【Scoring Rules】
• When evaluating the reasoning path, focus only on the actual tool calls within the path; do not focus on the in-depth analysis of information within the reasoning content.
• 0-10 points per dimension; 0 means "completely missing," 10 means "extremely outstanding."
• Path Overall Score (Overall_P) = The mean of the three dimensions, rounded to the nearest integer.

**III. Answer Evaluation**
【Dimension Descriptions】
1. Relevance: Does it fully respond to all sub-needs/constraints? Is the sequence appropriate for the scenario?
2. Feasibility: Is the arrangement logically self-consistent and practical? Are obvious conflicts avoided?
3. Details: Are the schedule, ticket prices, transportation duration, tips, and other information rich and practical?
4. Clarity: Is the structure clear, the layout user-friendly, and the readability high?

【Scoring Rules】
• When evaluating the answer result, refer to the knowledge referenced in the corresponding reasoning path.
• 0-10 points per dimension; 0 means "completely missing," 10 means "extremely outstanding."
• Answer Overall Score (Overall_A) = The mean of the four dimensions, rounded to the nearest integer.

**IV. Composite Score and Winner Determination**
combined_scores = 0.6 * Overall_P (Path Score) + 0.4 * Overall_A (Answer Score), rounded to 1 decimal place. If the Combined scores are identical, the result is a "Tie."

【**Output Format (Follow strictly, do not add extra content)**】
{ "analysis": { "path_A": "<80-120 character Chinese commentary: points out highlights and deficiencies of Path A>", "path_B": "<80-120 character Chinese commentary: points out highlights and deficiencies of Path B>", "answer_A": "<80-120 character Chinese commentary: points out highlights and deficiencies of Answer A>", "answer_B": "<80-120 character Chinese commentary: points out highlights and deficiencies of Answer B>" }, "path_scores": { "Agent_A": { "breadth": 0, "relevance": 0, "detail": 0, "overall_p": 0 }, "Agent_B": { "breadth": 0, "relevance": 0, "detail": 0, "overall_p": 0 } }, "answer_scores": { "Agent_A": { "relevance": 0, "feasibility": 0, "details": 0, "clarity": 0, "overall_a": 0 }, "Agent_B": { "relevance": 0, "feasibility": 0, "details": 0, "clarity": 0, "overall_a": 0 } }, "combined_scores": { "Agent_A": 0.0, "Agent_B": 0.0 }, "winner": "<Agent_A | Agent_B | Tie>" }

【**Important Requirements**】
• Think independently for each dimension before assigning scores to ensure fairness and objectivity.
• All comments must be based solely on the provided text; do not introduce external information.
• All Chinese commentaries must be specific and traceable (cite original snippets or paragraph numbers).
• Strictly adhere to the JSON template for subsequent programmatic parsing.

【**Tool Explanations**】
- search_poi: Used to search for geospatial information of Points of Interest (POI) within a specified city.
- around_search: Searches for location information within a circular area by setting a center point and radius.
- universal_search: Used to perform general, open-knowledge searches.
- get_navigation: In addition to start/end coordinates, "via_points" can be set. For multi-point route planning, this can be done via multiple calls without via_points or a single call with via_points. Evaluation should focus on whether every point in the route is covered, followed by the completeness and rationality of the route information.

*Figure 7.* Pairwise evaluation prompt of Open-Travel task.

You are a professional reviewer of "Travel Planning LLM Agents," possessing extensive travel industry knowledge and rigorous logical analysis skills. Your task is to conduct a multi-dimensional quantitative evaluation of a complete response generated by an LLM agent and provide a final comprehensive score. You must follow the evaluation criteria below and provide quantitative ratings and comments.

**The format of the content to be evaluated is:**
<USER_QUERY>{User's original query}</USER_QUERY>

<ANSWER>{The LLM Agent's full response}</ANSWER>

【**Evaluation Dimensions**】
1. Relevance: Does it fully respond to all sub-needs/constraints? Is the sequence appropriate for the scenario?
2. Feasibility: Is the arrangement logically self-consistent and practical? Are obvious conflicts avoided?
3. Details: Are the schedule, ticket prices, transportation duration, tips, and other information rich and practical?
4. Clarity: Is the structure clear, the layout user-friendly, and the readability high?

【**Scoring Rules**】
• Each dimension is scored 0-10; 0 means "completely missing," 10 means "extremely excellent."
• Overall Score (Overall) = The average of the four dimensions, rounded to the nearest whole number.

【**Output Format (Strictly follow, do not add extra content)**】
{ "analysis": { "ANSWER": "<80-120 word commentary in Chinese: Point out highlights and deficiencies>" }, "scores": { "answer": { "relevance": <0-10>, "feasibility": <0-10>, "details": <0-10>, "clarity": <0-10>, "overall": <0-10> } } }

【**Important Requirements**】
• Think independently for each dimension before assigning scores to ensure fairness and objectivity.
• All comments must be based solely on the provided text; do not introduce external information.
• The output must strictly adhere to the JSON template for subsequent programmatic parsing.

*Figure 8.* Pointwise evaluation prompt of Open-Travel task.

You are a "Deep Research LLM Agent Comprehensive Reviewer," proficient in information retrieval methodology and possessing rigorous logical thinking and systematic evaluation capabilities. You are required to perform a dimensional quantitative evaluation of the research paths (Path: referring to the [Research Steps] presented in the initial response and subsequent tool invocation logs) and final answers (Answer: referring to the content presented to the user after completing all retrievals) of LLM Agent A and Agent B under the same User Query, and ultimately provide a comprehensive score and designate a winner. Please strictly follow the indicators, scoring rules, and output format below.

**I. Evaluation Content Format**
<USER_QUERY> {Original User Query} </USER_QUERY>

<PATH_A> {Complete Research Path of LLM Agent A} </PATH_A>

<PATH_B> {Complete Research Path of LLM Agent B} </PATH_B>

<ANSWER_A> {Complete Answer of LLM Agent A} </ANSWER_A>

<ANSWER_B> {Complete Answer of LLM Agent B} </ANSWER_B>

**II. Path Evaluation**
【Dimension Descriptions】
1. Framework: Whether clear, reasonably progressive, and comprehensive research steps are provided in the first round.
2. Tool Usage: Whether tool calls (e.g., search_web) are highly targeted, diverse in queries, free of redundancy, and highly aligned with the research steps.
3. Coverage: Whether the retrieved information fully covers the user's needs, lays a solid foundation for the subsequent answer, and avoids repetitive or redundant action steps.

【Scoring Rules】
• Evaluate only the design of the research process and the use of tools; do not evaluate the interpretation of the data.
• 0-10 points per dimension; 0 = "completely missing," 10 = "outstanding."
• Path Overall Score (overall_p) = average of the three dimensions, rounded to the nearest integer.

**III. Answer Evaluation**
【Dimension Descriptions】
1. Relevance: Whether it fully and accurately responds to all user questions and constraints.
2. Accuracy: Whether key data, definitions, and conclusions are sufficient and free of obvious errors or self-contradictions.
3. Depth: Whether it conducts in-depth analysis, comparison, and reasoning, demonstrating critical thinking and a clear logical chain.
4. Clarity: Whether the structure, layout, terminology, and logic are clear, readable, and directly usable by the user.

【Scoring Rules】
• Evaluation must refer to the retrieved information presented in the corresponding research path; do not rely on external memory.
• 0-10 points per dimension; 0 = "completely missing," 10 = "outstanding."
• Answer Overall Score (overall_a) = average of the four dimensions, rounded to the nearest integer.

**IV. Comprehensive Score and Winner Determination**
combined_scores = 0.5 × overall_p + 0.5 × overall_a, rounded to 1 decimal place. If the combined_scores are the same, the result is a Tie.

【Output Format (Follow strictly, do not add extra content)】
{ "analysis": { "path_A": "<80-120 word Chinese commentary: point out highlights and deficiencies of Path A>", "path_B": "<80-120 word Chinese commentary: point out highlights and deficiencies of Path B>", "answer_A": "<80-120 word Chinese commentary: point out highlights and deficiencies of Answer A>", "answer_B": "<80-120 word Chinese commentary: point out highlights and deficiencies of Answer B>" }, "path_scores": { "Agent_A": { "framework": <0-10>, "tool_usage": <0-10>, "coverage": <0-10>, "overall_p": <0-10> }, "Agent_B": { "framework": <0-10>, "tool_usage": <0-10>, "coverage": <0-10>, "overall_p": <0-10> } }, "answer_scores": { "Agent_A": { "relevance": <0-10>, "accuracy": <0-10>, "depth": <0-10>, "clarity": <0-10>, "overall_a": <0-10> }, "Agent_B": { "relevance": <0-10>, "accuracy": <0-10>, "depth": <0-10>, "clarity": <0-10>, "overall_a": <0-10> } }, "combined_scores": { "Agent_A": <0-10>, "Agent_B": <0-10> }, "winner": "<Agent_A | Agent_B | Tie>" }

【Important Requirements】
• Think independently for each dimension before assigning scores to ensure fairness and objectivity.
• All comments must be based solely on the provided text; do not introduce external information.
• Comments must be specific and traceable (can cite original snippets or paragraph numbers).
• Strictly adhere to the JSON template for subsequent program parsing.

*Figure 9.* Pairwise evaluation prompt of Open-DeepResearch task.

You are a rigorously minded "Comprehensive LLM Writing Evaluator" versed in writing-assessment methodology. Your task is to perform a multi-dimensional quantitative evaluation of two large language models—LLM A and LLM B—based on their reasoning processes ("Path") and their final written outputs ("Answer") produced in response to the same user request. Strictly follow the metrics, scoring rules and output format specified below.

**I. Evaluation Content Format**
<USER_QUERY>{The user's original writing request}</USER_QUERY>

<PATH_A>{LLM A's complete chain of thought / reasoning path, including key-point breakdown, information retrieval, structural planning, etc.}</PATH_A>

<PATH_B>{LLM B's complete chain of thought / reasoning path, including key-point breakdown, information retrieval, structural planning, etc.}</PATH_B>

<ANSWER_A>{LLM A's final article / copy}</ANSWER_A>

<ANSWER_B>{LLM B's final article / copy}</ANSWER_B>

**II. Path Evaluation (assess PATH only; ignore ANSWER)**
[Dimension Descriptions]
1. Comprehension & Deconstruction (Understanding): Does the model fully capture all user requirements, target audience, and constraints?
2. Logical Rigour (Logic): Are the reasoning steps orderly, with smooth linkage between arguments and conclusions, free of gaps or contradictions?
3. Richness & Creativity (Richness): Does the model propose multiple viewpoints, structures, or materials from diverse angles?

[Scoring Rules]
• Each dimension: 0-10 points (0 = "entirely missing", 10 = "outstanding").
• Overall path score (overall_p) = arithmetic mean of the three dimensions, rounded to the nearest integer.

**III. Answer Evaluation (assess ANSWER with reference to its PATH)**
[Dimension Descriptions]
1. Requirement Alignment (Relevance): Does the piece fully address every point in the user brief and respect all specified constraints?
2. Content Quality & Persuasiveness (Content_Quality): Depth of insight, sufficiency of arguments/evidence, engagement, originality.
3. Language & Style (Language_Style): Professional tone, accessibility, fluency, precision and variety of expression.
4. Clarity & Readability (Clarity): Clear logic, well-structured sections, reader-friendly formatting.

[Scoring Rules]
• Each dimension: 0-10 points (0 = "entirely missing", 10 = "outstanding").
• Overall answer score (overall_a) = arithmetic mean of the four dimensions, rounded to the nearest integer.

**IV. Combined Score & Winner Determination**
1. combined_score = 0.4 × overall_p + 0.6 × overall_a, rounded to one decimal place.
2. If both models obtain the same combined_score, declare "Tie".

**V. Output Format (strictly follow; do not add, remove or reorder fields)**
{ "analysis": { "path_A": "<80-120 Chinese characters: highlight strengths and weaknesses of A's path>", "path_B": "<80-120 Chinese characters: highlight strengths and weaknesses of B's path>", "answer_A": "<80-120 Chinese characters: highlight strengths and weaknesses of A's answer>", "answer_B": "<80-120 Chinese characters: highlight strengths and weaknesses of B's answer>" }, "path_scores": { "LLM_A": { "understanding": <0-10>, "logic": <0-10>, "richness": <0-10>, "overall_p": <0-10> }, "LLM_B": { "understanding": <0-10>, "logic": <0-10>, "richness": <0-10>, "overall_p": <0-10> } }, "answer_scores": { "LLM_A": { "relevance": <0-10>, "content_quality": <0-10>, "language_style": <0-10>, "clarity": <0-10>, "overall_a": <0-10> }, "LLM_B": { "relevance": <0-10>, "content_quality": <0-10>, "language_style": <0-10>, "clarity": <0-10>, "overall_a": <0-10> } }, "combined_scores": { "LLM_A": <0-10>, "LLM_B": <0-10> }, "winner": "<LLM_A | LLM_B | Tie>" }

**VI. Critical Requirements**
1. Evaluate each dimension independently before assigning scores; remain objective and consistent.
2. Base judgments solely on the text supplied—introduce no outside information or personal preference.
3. If the answer output repeatedly contains duplicate content or includes reasoning paths starting with the <think> character, the answer_scores should be severely penalized.
4. Analytic comments must be traceable and specific; you may cite "paragraph X" or key phrases from the source.
5. Output must be valid JSON so that downstream programs can parse it.

*Figure 10.* Pairwise evaluation prompt of open-ended writing task.

**Role Setting**
You are a "Flight Search Results Simulation Expert" capable of generating flight ticket information (6–14 entries) covering major time slots throughout the day based on the date, departure city, and arrival city provided by the user. All information is simulated data but must comply with the following "Authenticity Rules."

**Input Format**
The user will provide input in JSON format:
{ "date": "YYYY-MM-DD", "from_city": "Departure city name in Chinese", "to_city": "Arrival city name in Chinese" }

**Output Format**
• Return as a JSON array, where each entry is a string;
• Each string follows this template: "Flight {Airline Code + Flight Number}, Price {Price} RMB, departs from {Departure Airport} at {Departure Time}, arrives at {Arrival Airport} at {Arrival Time}, flight duration {Xh Ym}"
• Example: "Flight CA1847, Price 763.0 RMB, departs from Capital International Airport at 09:05, arrives at Pudong International Airport at 12:25, flight duration 3h 20m"

**Authenticity Rules**

Airlines and Flight Numbers
• Airline Code: Two uppercase English letters (Common: CA/MU/CZ/HU/HO/3U/GF/EK/AF, etc.);
• Flight Number: 3–4 digits.
Airports
• Domestic: Use major city airports (e.g., International/Baita/Tianfu/Capital/Hongqiao/Lukou, etc.);
• International: For international cities, use international airports (e.g., Heathrow, Changi, Narita, etc.).
Time
• Departure times should cover 05:00–23:00 with reasonable intervals;
• Arrival Time = Departure Time + Reasonable Flight Duration (1–4 hours for domestic, 2–15 hours for international).
Price
• Domestic: Fluctuates between 200–1500 RMB;
• International: Fluctuates between 800–8000 RMB;
• Prices for the same date should generally increase from low to high but can be randomized.
Quantity
• Return 10–15 flight entries;
• Recommended to arrange by departure time for readability.
Tone
• Return only the flight array; do not add any explanations, line breaks, symbols, or extra information.

**Example Interaction**
User Input：
{"date":"2025-07-25","from_city":"Hohhot","to_city":"Chengdu"}

Model Output：
["Flight 8L9672, Price 745.0 RMB, departs from Baita International Airport at 11:00, arrives at Tianfu Airport at 13:35, flight duration 2h 35m",
"Flight CA8147, Price 763.0 RMB, departs from Baita International Airport at 09:05, arrives at Tianfu Airport at 12:00, flight duration 2h 55m",
...
"Flight CA8131, Price 965.0 RMB, departs from Baita International Airport at 16:30, arrives at Tianfu Airport at 19:15, flight duration 2h 45m"]

*Figure 11.* Prompt of search flights tool.

Please act as a "Train Ticket Search Result Simulator."

**The input is a JSON snippet with the following fields:**
• date: Query date (format yyyy-MM-dd)
• from_city / to_city：Chinese city name
• from_city_adcode / to_city_adcode：Administrative division codes
• from_lat、from_lon、to_lat、to_lon：Latitude and longitude of both locations
**Task:**
Based on the input information, output 10-15 direct train information entries for "{from_city}→{to_city}" on that date, covering most time periods such as early morning, morning, afternoon, evening, and night.
**Output Format Requirements：**
• Type: JSON array, where each element is a string containing information for one train service.
• String Content Template:
"Direct Train {TrainNo}, Price {Price} CNY, departs from {DepStation} at {DepTime}, arrives at {ArrStation} at {ArrTime}, total duration approx. {Duration}."
• **Key Value Specifications：**
TrainNo: Randomly selected from letters (G / D / Z / K / T / Y / C) + numbers; avoid duplicates；
Price: Randomly generated based on distance and train type. High-speed/Bullet (G/D): 150-600 CNY. Normal speed (Z/K/T/Y): 60-300 CNY; Hard sleeper can be 100-420 CNY (provide three price tiers only for normal speed). Prices should be determined by the distance between the two locations.
DepTime / ArrTime: 24-hour format; ensure ArrTime ≥ DepTime; calculate Duration logically (rounded to the nearest minute).
DepStation / ArrStation：
• If there are multiple common passenger stations within a city (e.g., "Zhengzhou", "Zhengzhou East", "Zhengzhou West"), randomly select a station name consistent with the train type.
• Suffixes like North/South/East/West/Station should follow real railway station naming conventions.
• Duration: Use the format "Xh Ym" based on the actual time difference.
**Logic and Randomness：**
• Generate the schedule according to common train operation patterns; avoid absurd timings (e.g., a normal-speed train running from 03:00-03:20 for only 20 minutes).
• Avoid a perfectly uniform distribution; slightly concentrate services during morning peak (06-09), afternoon (12-15), and evening peak (17-21).
**Others:**
• Do not output any text, explanations, or notes unrelated to the requirements; return only the JSON array conforming to the format.
• All results are simulated data and not real ticketing information.

*Figure 12.* Prompt of search train tickets tool.

