# OpenReview forum: "ArenaRL: Scaling RL for Open-Ended Agents via Tournament-based Relative Ranking"
_ICML.cc/2026/Conference — ICML 2026 regular_

### Official Review · Reviewer_tQ9S · 2026-02-20

**Soundness:** 2
**Presentation:** 2
**Significance:** 2
**Originality:** 3
**Overall Recommendation:** 5
**Confidence:** 4

**Summary:**

This paper tackles the discriminative collapse issue in reinforcement learning (RL) for open-ended LLM agent tasks, where traditional pointwise scalar reward scoring fails due to low intra-group variance and LLM judge noise. To address this, the authors propose ArenaRL, a novel RL paradigm that leverages intra-group relative ranking instead of pointwise rewards. ArenaRL features a process-aware pairwise evaluation mechanism and a seeded single-elimination tournament (with linear $O(N)$ complexity) for high-fidelity ranking, converting rankings into normalized advantage signals to drive stable policy optimization. Additionally, the paper introduces two comprehensive benchmarks (Open-Travel and Open-DeepResearch) for open-ended agent evaluation.

While the proposed method achieves strong experimental results, since the paper's main contributions lie in the introduction of two benchmarks and the design of a ranking method (issues arising from direct scoring by LLM judges have been discussed in subsequent work, and the idea of using comparison to replace direct LLM judge scoring was not originally proposed in this paper), more explanations are needed to justify the novelty and significance of the method design in this work.

**Compliance With Llm Reviewing Policy:**

Affirmed.

**Final Justification:**

Thank you for your efforts during the rebuttal stage. The additional experiments and clarifications you have provided have satisfactorily addressed the questions and suggestions I raised in my initial review.

Accordingly, I am willing to raise my score from 3 to 5. I hope the authors will incorporate the corresponding revisions into the final version of the manuscript.

**Key Questions For Authors:**

My concerns align with the points outlined in the Weaknesses section. I hope the authors can address my doubts regarding the algorithm's contribution through relevant discussions and supplementary analyses.

**Limitations:**

Yes

**Strengths And Weaknesses:**

### Strengths
1. The two benchmark datasets (Open-Travel and Open-DeepResearch) constructed in this paper fill the gap in the current field of open-ended agents, where there is a lack of complete training-evaluation full-process benchmarks, and provide standardized and reproducible experimental infrastructure for research in this direction.
2. The proposed ArenaRL algorithm significantly outperforms mainstream baseline methods (including classic RL algorithms and closed-source large language models) on three tasks: travel planning, deep research, and open-ended writing.
3. The paper has a clear structure, and the descriptions of core methods (process-aware pairwise evaluation, seeded single-elimination tournament) and benchmark construction processes are detailed and easy to understand, complying with the presentation norms of academic papers.

### Weaknesses
1. As stated by the authors in the related work section, the core idea of "using comparison to replace direct scoring by LLM judges" is not original to this paper. Its main methodological innovation focuses on the design of a specific ranking mechanism, leading to insufficient demonstration of the novelty and significance of the overall method, and failing to highlight the essential differences from existing comparison-based evaluation ideas.
2. The experiments lack direct comparison with two existing comparison-based evaluation methods: Writing-Zero and Pref-GRPO. Although the theoretical computational complexity of these two methods is higher than that of ArenaRL, verifying whether ArenaRL achieves a better trade-off between performance and efficiency through experiments is an important basis for supporting its advantages. The current lack of such comparison results in incomplete argumentation.
3. The core innovation of this paper at the algorithm level is the seeded single-elimination ranking method, but the experiments only compare it with basic ranking topologies such as round-robin and anchor-based ranking, without incorporating advanced ranking methods existing in the field. The lack of experimental comparison with such methods makes it difficult to fully verify the advancement of the proposed ranking mechanism on this task.
4. In Table 2, on the Compare and 1-Day subtasks of the Open-Travel benchmark, the performance of the seeded single-elimination system is significantly inferior to that of the double-elimination system, with only a slight advantage in the overall mean value. This phenomenon has not been explained in detail by the authors, which weakens the persuasiveness of the conclusion that "seeded single-elimination is the optimal topology". Meanwhile, the experiments do not supplement the performance of different tournament topologies on the Open-DeepResearch benchmark, leading to insufficient comprehensive support for the generality and robustness of topology selection.

---

> ### Author Rebuttal · Authors · 2026-03-31
>
> Dear Reviewer tQ9S:
>
> We sincerely thank you for the valuable comments! Your comments are crucial for us to improve our work. We will address your concerns point by point.
>
> >**Q1:** Difference from existing comparison-based methods
>
> **A1:** ArenaRL differs from prior comparison-based methods (e.g. Writing-Zero, Pref-GRPO) in two main aspects:
>
> **(1) Task domain and evaluation granularity:** Writing-Zero and Pref-GRPO target single-turn tasks (e.g., creative writing or text-to-image) with outcome-only evaluation. In contrast, ArenaRL addresses long-horizon agentic tasks requiring multi-turn tool use and reasoning. It introduce a process-aware evaluation mechanism that evaluates both intermediate tool-use and final outcome via multi-dimensional rubrics, providing fine-grained supervision for complex trajectories.
>
> **(2) Algorithmic efficiency and signal quality:** Existing methods such as Writing-Zero rely on coarse binary win–loss signals and compete only against random references, while Pref-GRPO estimates each sample’s win rate through exhaustive $O(N^2)$ pairwise comparisons. ArenaRL achieves a good trade-off by introducing a seeded single-elimination tournament, reducing computational complexity to $O(N)$ while still providing high-fidelity, ranking-based advantage signals. Despite the lower cost, ArenaRL maintains strong ranking consistency, achieving a Spearman correlation of ρ = 0.961 with round-robin rankings.
>
>
> >**Q2:** Experiments lack direct comparison with Writing-Zero and Pref-GRPO
>
> **A2:** Thank you for raising this important feedback. We added experiments for both methods on the Open-Travel task. As shown below, ArenaRL outperforms both methods. We attribute this to the fact that without process-aware evaluation, Writing-Zero and Pref-GRPO struggle to provide high-quality advantage signals for multi-turn agentic task.
>
> |Method|Complexity|Direction|Search|Compare|1-Day|M-Day|Mean|
> |:---|:---:|:---:|:---:|:---:|:---:|:---:|:---:|
> |SFT|—|10.6|29.7|14.1|20.4|7.1|16.4|
> |Writing-Zero|$O(N)$|17.3|43.2|20.8|33.1|14.5|25.8|
> |Pref-GRPO|$O(N^2)$|25.7|50.4|27.5|41.2|16.4|32.2|
> |ArenaRL|$O(N)$|32.1|66.1|31.7|58.0|21.0|41.8|
>
> >**Q3:** Missing comparison with advanced ranking methods.
>
> **A3:** To further validate our tournament design, we implemented Glicko-2 [1] (a Bayesian rating system) and Copeland [2] (a voting-based Condorcet method) on the Open-Travel task. As shown below, ArenaRL achieves the best performance while retaining $O(N)$ complexity. The Glicko-2 relies on long-term interaction history to update confidence and may suffer from unstable signals under limited comparisons. Copeland has $O(N^2)$ complexity, but performs slightly worse than our proposed seeded single-elimination method.
>
> |Method|Complexity|Direction|Search|Compare|1-Day|M-Day|Mean|
> |:---|:---:|:---:|:---:|:---:|:---:|:---:|:---:|
> |SFT|—|10.6|29.7|14.1|20.4|7.1|16.4|
> |Glicko-2|$O(N)$|26.8|55.4|27.3|54.2|16.4|36.0|
> |Copeland|$O(N^2)$|30.1|64.2|33.9|53.7|18.4|40.1|
> |ArenaRL|$O(N)$|32.1|66.1|31.7|58.0|21.0|41.8|
>
> >**Q4:** Why seeded single-elimination underperforms double-elimination on Compare and 1-Day subtasks?
>
> **A4:** We appreciate this insightful observation. Compared with more complex tasks such as M-Day or Search, Compare and 1-Day have smaller solution spaces and fewer constraints. In this context, the anchor generated by greedy decoding in seeded single-elimination may occasionally converge to a local optimum, slightly affecting ranking quality. Double-elimination provides eliminated trajectories a second chance, this fault tolerance ensures high-quality rankings in simple tasks. However, for complex tasks, its ranking quality is weaker than seeded single-elimination (e.g., on M-Day, seeded single-elimination achieves 18.1%, clearly higher than the 12.3% of double-elimination).
>
> >**Q5:** Performance of different tournament topologies on Open-DeepResearch
>
> **A5:** To further evaluate the various topologies, we compared five tournament topologies on the Open-DeepResearch benchmark. As shown below, the Seeded single-elimination achieves superior performance, highlighting its advantages.
>
> |Tournament Topology|Frm.|Tool|Cov.|Rel.|Acc.|Dep.|Cla.|Mean|Val.(%)|
> |:---|:---:|:---:|:---:|:---:|:---:|:---:|:---:|:---:|:---:|
> |SFT|14.1|20.3|23.4|14.1|15.6|15.6|14.1|16.7|32.0|
> |Anchor-Based|48.5|55.2|56.1|46.4|49.5|44.3|47.8|49.7|89.0|
> |Swiss-System|55.2|68.6|72.3|55.2|49.5|49.7|55.2|58.0|86.0|
> |Double-Elimination|50.8|66.4|69.2|54.9|48.9|53.6|53.6|56.8|94.0|
> |Seeded Single-Elimination|62.6|77.3|78.8|57.1|55.6|57.1|61.6|64.3|99.0|
> |Round-Robin|65.4|79.1|80.6|55.7|57.3|61.2|63.1|66.1|98.0|
>
> [1] Daynauth, et al. "Ranking unraveled: Recipes for llm rankings in head-to-head ai combat." ACL 2025.
>
> [2] Van Elst A,  et al. Decentralized Ranking Aggregation: Gossip Algorithms for Borda and Copeland Consensus[J]. arXiv:2602.22847, 2026.
>
> Thank you once again for your time and valuable feedback!

---

> > ### Author Rebuttal · Reviewer_tQ9S · 2026-04-03
> >
> > Thank you for your careful and detailed response, as well as the supplementary experiments you provided. Your efforts to address my concerns are highly appreciated, and the additional experiments have partially resolved several of my previous questions. However, there are still two key issues that require further clarification, which I believe are crucial for evaluating the validity and generality of your work:
> > (1) Regarding my Q4: The performance degradation (from 33.7 to 22.9 and 39.9 to 34.9) can hardly be described as a "slight" loss. The explanation you provided fails to fully convince me that this is a reasonable trade-off; instead, it seems more like an approach that sacrifices performance on simple tasks to prioritize difficult cases. I truly appreciate your sincere response, but this issue is particularly important—especially since Open-Travel is the benchmark proposed by your team. A significant performance drop on simple subtasks of your own benchmark may raise concerns about the overall robustness and practicality of the proposed seeded single-elimination tournament design.
> >
> > (2) You have not provided the corresponding results of Writing-Zero and Pref-GRPO on the "Performance Comparison on Open-Ended Writing Task Across Three Public Benchmarks". In fact, all supplementary experiments are conducted on the two benchmarks proposed by your team. To verify that the performance improvement of your method on your self-proposed benchmarks is not achieved at the cost of generalization ability, I kindly request you to provide the comparison results of these two methods (Writing-Zero and Pref-GRPO) on public datasets. This will help confirm that your method has good generalization beyond the benchmarks you constructed.
> >
> > If you can provide the corresponding proofs and explanations for the above two points, I will re-evaluate the current version of your work based on your responses and supplementary experiments.

---

> > > ### Author Response · Authors · 2026-04-06
> > >
> > > Dear Reviewer tQ9S:
> > >
> > > We sincerely appreciate your time and expertise in reviewing our work. We apologize that our previous explanation did not fully address your concerns and will now offer a clearer response to your questions.
> > >
> > > >**Q1:** Why does seeded single-elimination underperform double-elimination on the Compare and 1-Day subtasks?
> > >
> > > **A1:** Thank you for the careful observation. We agree that describing the performance degradation as “slight” in our previous response was inaccurate.
> > >
> > > To more comprehensively evaluate the robustness and practicality of the proposed seeded single-elimination tournament, we further compared the two ranking strategies under larger group sizes (N=16 and N=32). The results are shown in Table 1, where N=8 corresponds to Table 2 in the paper. The results indicate that the noticeable disadvantage of seeded single-elimination on the Compare and 1-Day subtasks appears only at the small group size (N=8). **As the group size increases, the advantages of the seeded single-elimination gradually emerge and it eventually achieves the best performance across all subtasks.**
> > >
> > > This behavior mainly stems from the random matching mechanism used in double-elimination. When N=8, random pairings can still provide effective ranking signals for relatively simple tasks. However, as group size N grows, the policy model produces more top-tier trajectories within each group. Random matching then increases the probability that these top-tier trajectories encounter each other early and are prematurely eliminated. Even with a losers’ bracket, repeated random matchups within that bracket make it difficult to reliably distinguish top-tier trajectories, introducing noise into the final advantage signal.
> > >
> > > In contrast, seeded single-elimination scales more effectively. The method first pre-ranks the N trajectories using greedy anchors, and then applies serpentine grouping scheme (e.g., seed 1 vs. seed N) in the elimination stage. This structured topology reduces early elimination among top-tier trajectories, allowing the final advantage signal to be derived from more informative comparisons.
> > >
> > > Table 1: Performance comparison across different group sizes N on Open-Travel benchmark
> > >
> > > |N|Method|Direction|Search|Compare|1-Day|M-Day|Mean|
> > > |:---:|:---|:---:|:---:|:---:|:---:|:---:|:---:|
> > > |8|Double-Elimination|12.6|52.4|**33.7**|**39.9**|12.3|30.2|
> > > |8|**Seeded Single-Elimination**|**16.9**|**69.9**|22.9|34.9|**18.1**|**32.5**|
> > > |- |- |- |- |- |- |- |
> > > |16|Double-Elimination|18.3|56.2|**36.0**|49.5|17.6|35.5|
> > > |16|**Seeded Single-Elimination**|**32.1**|**66.1**|31.7|**58.0**|**21.0**|**41.8**|
> > > |- |- |- |- |- |- |- |
> > > |32|Double-Elimination|24.7|58.8|35.4|53.6|18.5|38.2|
> > > |32|**Seeded Single-Elimination**|**37.1**|**71.9**|**37.5**|**63.1**|**24.7**|**46.9**|
> > >
> > > >**Q2:** Comparison with Writing-Zero and Pref-GRPO on open-ended writing benchmarks
> > >
> > > **A2:** To further evaluate the generalization ability of ArenaRL, we conducted additional experiments on three public open-ended writing benchmarks: WritingBench (WB), HelloBench (HB), and LongBench-write (LB), comparing with Writing-Zero and Pref-GRPO.
> > >
> > > Table 2: Performance comparison on three open-ended writing benchmarks
> > >
> > > |Method|WB-A|WB-B|WB-C|WB-D|WB-E|WB-F|HB-Summ.|HB-Heur.|LB-quality|Mean|
> > > |:---|:---:|:---:|:---:|:---:|:---:|:---:|:---:|:---:|:---:|:---:|
> > > |SFT|70.7|69.4|67.9|63.7|69.7|70.6|63.4|82.4|85.5|71.5|
> > > |Writing-Zero|73.3|73.8|69.8|68.2|72.5|71.8|68.1|85.2|88.5|74.6|
> > > |Pref-GRPO|74.5|73.8|73.2|72.4|75.1|74.6|70.3|90.1|91.0|77.2|
> > > |**ArenaRL**|**78.1**|**77.7**|**77.6**|**75.0**|**79.4**|**77.2**|**73.8**|**91.3**|**93.8**|**80.4**|
> > >
> > > ArenaRL outperforms both baselines on all nine evaluation dimensions, indicating that the proposed process-aware tournament ranking method not only benefits open-ended agent tasks but also provides more effective advantage signals for open-ended writing.
> > >
> > > We deeply value your expertise and hope that our responses and the new results have addressed your concerns. We would be very grateful to know if you have any remaining questions or if there's anything that requires further clarification. If our rebuttal has successfully resolved your initial concerns, we would be deeply appreciative if you would consider re-evaluating our paper.
> > >
> > > Many thanks for your constructive comments, time, and patience.
> > >
> > > Best regards and thanks,
> > >
> > > The Authors

---

### Official Review · Reviewer_3Yiu · 2026-03-11

**Soundness:** 3
**Presentation:** 4
**Significance:** 3
**Originality:** 2
**Overall Recommendation:** 5
**Confidence:** 3

**Summary:**

This paper investigated reinforcement learning for open-ended agents, and propose several tournament-based ranking system to enrich the reward signal for policy optimization. The authors also propose benchmarks for evaluating open-ended agents. The proposed method ArenaRL achieves high precision advantage estimation and demonstrate superior performance on these tasks compared to other RL methods as well as state-of-the-art close-source models.

**Compliance With Llm Reviewing Policy:**

Affirmed.

**Key Questions For Authors:**

- It's reasonable to leverage more reward signals for policy optimization. However, the proposed pairwise evaluation to obtain rich reward/advantage signals seem related to **general preference optimization**, where pairwise preference scores are obtained using ranking LLMs or LLM-as-a-duge, and some works in this family can also have low complexity such as [1] . How do the authors compare your method with this family? Or maybe the question becomes do you think preference-based RL can be beneficial here? I understand preference-base reward signal might be limited here compared to multiple rubrics.

- The authors mentioned this is a long-horizon setting, but what are average horizons of the two proposed benchmarks (Open-Travel and Open-DeepResearch? It'd be helpful to have some statistics about horizon of the optimal trajectory, or some average horizon.

**Limitations:**

- Is there any other rank-based Policy Optimization algorithm? The related work seems not covering them. I believe worth covering or even comparing with in the experiments.
- I think the related work section of RL with LLMs is a bit limited. Literatures of obtaining reward signals from (pariwise) comparison seems not enough. Besides, Pref-grpo is for text-to-image tasks. Works from general preference optimization is worth selectively covering [1,2,3,4,5].


[1] Zhang, Yuheng, et al. "Iterative nash policy optimization: Aligning llms with general preferences via no-regret learning." arXiv preprint arXiv:2407.00617 (2024).

[2] Munos, Rémi, et al. "Nash learning from human feedback." Forty-first International Conference on Machine Learning. 2024.

[3] Zhang, Yifan, et al. "General preference modeling with preference representations for aligning language models." (2024).

[4] Wu, Yue, et al. "Self-play preference optimization for language model alignment." arXiv preprint arXiv:2405.00675 (2024).

[5] Tang, Xiaohang, et al. "RSPO: Regularized Self-Play Alignment of Large Language Models." arXiv preprint arXiv:2503.00030 (2025).

**Strengths And Weaknesses:**

- Presentation: The paper is well-written, proposing a widely existed problem about the noisy reward for open-ended agents. And then a tournament-based ranking system is proposed to extract the reward for conducting RL. The motivation, structure and the methods are clearly presented.
- Soundness: The experimental results are solid. There is a significant improvement over the baseline methods and baseline models.
- Significance: The open-ended tasks are important to solve. It advances capabilities of LLMs, and provides new evaluation benchmarks in this field.
- Originality: See Limitation.

---

> ### Author Rebuttal · Authors · 2026-03-31
>
> Dear Reviewer 3Yiu:
>
> We sincerely thank you for the valuable comments! Your comments are crucial for us to improve our work. We will address your concerns point by point.
>
> >**Q1:** Difference between ArenaRL and preference optimization methods, and applicability to open-ended agent tasks.
>
> **A1:** Thank you for highlighting related general preference optimization works [1–5]. ArenaRL differs from these methods primarily in two aspects:
>
> **(1) Evaluation granularity:** Traditional preference optimization methods typically focuses on single-turn response generation tasks, where preferences are defined solely by the final output. However, open-ended agent tasks involve long-horizon trajectories where intermediate steps directly influence the final outcome. ArenaRL introduces a process-aware evaluation mechanism using multi-level rubrics. By jointly assessing reasoning coherence, tool-use accuracy, and final answer reliability, ArenaRL provides fine-grained supervision across the entire trajectory.
>
> **(2) Optimization objective:** The preference optimization methods typically rely on no-regret learning or self-play to gradually converge to the Nash equilibrium, requiring multiple iterations with complexity roughly O(T·N). ArenaRL instead adopts online RL training based on intra-group relative ranking. The constructed seeded single-elimination scheme performs group ranking in O(N) complexity without iterative convergence, and the resulting discrete ranks are mapped into stable advantage signals that integrate naturally with the GRPO-based optimization framework.
>
> >**Q2:** Statistics on the average horizons of the two proposed benchmarks.
>
> **A2:** We provide the interaction horizon statistics for both benchmarks below:
>
> Table 1: Open-Travel Statistics
>
> |Metric|Direction|Search|Compare|1-Day|M-Day|Mean|
> |:---|:---:|:---:|:---:|:---:|:---:|:---:|
> |Avg.Turns|10.42|5.66|8.10|7.34|7.52|7.81|
> |Avg.Tokens|9,653.18|5,697.70|7,725.40|14,114.54|26,993.02|12,836.77|
>
> Open-DeepResearch statistics: Average turns: 9.93, Average tokens: 24,738.69
>
> These results indicate that both benchmarks involve long-horizon interactions. In particular, Open-DeepResearch trajectories typically exceed 20k tokens and nearly 10 dialogue turns, reflecting the long-term nature of these tasks.
>
> >**Q3:** Comparison with other ranking-based policy optimization algorithms.
>
> **A3:** Thanks for the suggestion. We categorize related ranking-based methods and compare them with ArenaRL as follows:
>
> ● PRO [6] and RCPO [7]: These methods model global ranking or use discrete choice theory (e.g., Top-K rankings) to optimize policy models to align with human preferences.
>
> ● λ-DPO [8]: This method performs listwise preference optimization with simplex-weighted objectives, enabling dynamic alignment with multiple preference dimensions.
>
> **Key Distinction**: These methods primarily focus on offline alignment with static datasets. In contrast, ArenaRL operates in an online RL setting. It generates stable, ranking-based advantage signals during active exploration, which is more effective for open-ended agent tasks.
>
> We reproduced PRO [6] and λ-DPO [8] on the Open-Travel benchmark, with the results further validating the effectiveness of ArenaRL.
>
> Table 2: Performance of Other Ranking-Based Policy Optimization Algorithms
>
> |Method|Direction|Search|Compare|1-Day|M-Day|Mean|
> |:---|:---:|:---:|:---:|:---:|:---:|:---:|
> |SFT|10.6|29.7|14.1|20.4|7.1|16.4|
> |PRO|28.3|45.1|18.5|43.2|15.8|30.2|
> |λ-DPO|24.1|38.4|18.7|35.6|13.6|26.1|
> |ArenaRL|32.1|66.1|31.7|58.0|21.0|41.8|
>
> >**Q4:** Insufficient discussion of related work on RL for LLMs.
>
> **A4:** Thank you for your valuable feedback. In the revision, we will expand the related work section to include general preference optimization methods [1–5], ranking-based policy optimization approaches [6–8], and rubric-based RL training methods [9–10].
>
> [1] Iterative nash policy optimization: Aligning llms with general preferences via no-regret learning. arXiv preprint arXiv:2407.00617.
>
> [2] Nash learning from human feedback. ICML. 2024.
>
> [3] General preference modeling with preference representations for aligning language models. (2024).
>
> [4] Self-play preference optimization for language model alignment. arXiv:2405.00675.
>
> [5] RSPO: Regularized Self-Play Alignment of Large Language Models. arXiv:2503.00030.
>
> [6] Preference Ranking Optimization for Human Alignment, AAAI 2024.
>
> [7] Beyond Pairwise: Empowering LLM Alignment With Ranked Choice Modeling. ICLR 2026.
>
> [8] Multi-Preference Lambda-weighted Listwise DPO for Dynamic Preference Alignment. arXiv: 2506.19780.
>
> [9] DR Tulu: Reinforcement Learning with Evolving Rubrics for Deep Research.  arXiv:2511.19399.
>
> [10] Advancedif: Rubric-based benchmarking and reinforcement learning for advancing llm instruction following. arXiv:2511.10507.
>
> Thank you once again for your time and valuable feedback!

---

> > ### Author Rebuttal · Reviewer_3Yiu · 2026-04-03
> >
> > Fully resolved.

---

> > > ### Author Response · Authors · 2026-04-03
> > >
> > > Dear Reviewer 3Yiu:
> > >
> > > We sincerely appreciate your positive feedback and are very glad to hear that our explanations have fully addressed your questions and concerns.
> > >
> > > We truly value the perspective you brought to our work, which has helped us improve our manuscript.
> > >
> > > Many thanks for your constructive comments, time, and patience.
> > >
> > > Best regards and thanks,
> > >
> > > The Authors

---

### Official Review · Reviewer_jUVH · 2026-03-13

**Soundness:** 3
**Presentation:** 2
**Significance:** 3
**Originality:** 3
**Overall Recommendation:** 4
**Confidence:** 4

**Summary:**

This paper proposes ArenaRL, an online RL method  that uses a tournament-based relative ranking scheme to generate more informative reward signals by increasing the intra-variance between candidate samples. Instead of scoring each trajectory independently, ArenaRL ranks trajectories within each generated group using a process-aware pairwise evaluation that considers not only the final outcome but also intermediate reasoning steps and tool use. ArenaRL is evaluated on two proposed benchmarks.

**Compliance With Llm Reviewing Policy:**

Affirmed.

**Ethical Review Concerns:**

Limitations are not discussed.

**Final Justification:**

The authors provided a strong rebuttal, adding additional baselines and datasets that addressed my main concerns. However, I believe they should still include a more datasets and include a training/inference time comparison between SWIRL and ArenaRL. I expect their approach to be more computationally expensive given the $O(N)$ complexity of the tournament scoring mechanism – that may not justify the performance improvement.

Due to the strong rebuttal and the empirical results provided during the discussion phase, I am raising my score to a Weak Accept.

**Key Questions For Authors:**

- What is the training-time computational cost of ArenaRL compared to pointwise baselines such as GRPO, in terms of wall-clock time and token usage?
- Why is the round-robin comparison described as “intractable,” given the relatively small group sizes used in the experiments?
- Can the authors report results for the standard RL baselines when equipped with process-aware rewards, to ensure a fair comparison with ArenaRL?
- Can the authors clarify why the proposed datasets evaluate open-ended tasks instead of multi-hop reasoning tasks?

**Limitations:**

Limitations are not discussed.

**Strengths And Weaknesses:**

Strengths:
- The idea of replacing pointwise scalar rewards with a tournament-based pairwise comparison scheme is interesting.
- The algorithmic design—reducing the comparison complexity to $O(N)$ through a seeded single-elimination tournament—is well-executed.

Weaknesses:
- Limited evaluation scope. While the introduction of the new benchmarks is important, the experimental setup relies  on these newly proposed datasets. Evaluating the method on additional established multi-hop benchmarks would strengthen the empirical evaluation.
- Missing process-reward baselines. The baseline comparisons may not be fair. The standard RL baselines (e.g., GRPO, GSPO) use pointwise rewards based on the final answer, whereas ArenaRL uses a process-aware evaluation that also considers intermediate reasoning steps. This makes it difficult to isolate whether the observed improvements come from the ranking mechanism itself or from the process-reward mechanism.
- Unclear computational overhead. Although the tournament design reduces comparisons relative to a round-robin scheme, the paper does not clearly report the training-time cost, such as wall-clock time, token consumption, or judge inference overhead compared to the baselines.
- Overstated complexity claim. The paper describes the O(N²) round-robin comparison as “intractable.” Given that group sizes used in RL training are typically small (e.g., N = 8–16), this characterization seems overstated. A more accurate description would be that it is computationally expensive rather than intractable.
- Presentation. The proposed datasets are on multi-hop reasoning instead of open-endedness.

---

> ### Author Rebuttal · Authors · 2026-03-31
>
> Dear Reviewer jUVH:
>
> We sincerely thank you for the valuable comments! Your comments are crucial for us to improve our work. We will address your concerns point by point.
>
> >**Q1:** Why does the proposed datasets evaluate open-ended tasks rather than multi-hop reasoning tasks?
>
> **A1:** Although our datasets involves multi-hop reasoning, it differs from standard multi-hop tasks such as HotpotQA [1]. These traditional multi-hop reasoning tasks typically have a single verifiable ground-truth answer, enabling the design of explicit reward functions based on answer matching. In contrast, Open-Travel and Open-DeepResearch operate in a large and subjective solution space where outputs are full travel plans or research reports **without a unique gold answer**. As a result, constructing verifiable reward functions is difficult, making conventional methods such as GRPO hard to apply directly. We will clarify this distinction between open-ended agent tasks and standard multi-hop reasoning tasks in the revision.
>
> >**Q2:** Evaluation on additional established multi-hop benchmarks.
>
> **A2:** Thank you for raising this critical question. To further evaluate ArenaRL, we conducted additional experiments on DeepResearch Bench [2]:
>
> | Method | Overall | Comp. | Insight | Inst. | Read. |
> | :--- | :---: | :---: | :---: | :---: | :---: |
> | SFT | 26.2 | 21.6 | 25.7 | 31.8 | 29.2 |
> | GRPO | 28.6 | 25.1 | 26.8 | 33.5 | 33.7 |
> | ArenaRL | 39.2 | 38.4 | 35.9 | 43.7 | 42.5 |
>
> ArenaRL substantially outperforms GRPO on DeepResearch Bench, further validating its effectiveness. In addition, as shown in Table 4 of our paper, ArenaRL consistently surpasses GRPO, GSPO, and Reinforce++ on three public open-ended writing benchmarks, maintaining a stable advantage across tasks.
> >**Q3:** Computational cost of ArenaRL compared with GRPO and other baselines.
>
> **A3:** In a uniform RL setting with group number K = 8 and group size N = 16, we measured the runtime per RL step, LLM token usage, and LLM evaluation count on the Open-Travel task:
> | Method | Training Time | Tokens (M) | LLM Evaluation Count | Performance |
> | :--- | :---: | :---: | :---: | :---: |
> | GRPO | 57 min | 0.29 | 128 | 16.4% |
> | Round-robin | 2 h 27 min | 5.66 | 960 | 40.9% |
> | ArenaRL | 1 h 13 min | 1.41 | 240 | 41.8% |
>
> Although GRPO uses the fewest tokens, its performance does not improve relative to that of the SFT model (16.4%). ArenaRL achieves much higher performance with a moderate increase in tokens, even slightly outperforming the round-robin. Moreover, most additional tokens in ArenaRL come from the input side, where two reasoning trajectories are input for each evaluation. Under typical LLM API pricing, input tokens are much cheaper than output tokens, so the overall cost increase is limited.
>
> >**Q4:** Why is the round-robin described as "intractable"?
>
> **A4:** Thank you for pointing this out. We agree that describing round-robin as “intractable” is imprecise, and will revise it to “computationally expensive”  in the revision. Additionally, as shown in the table below (derived from Fig. 4(a) of our paper), ArenaRL performance improves as the group size N increases. In contrast to ArenaRL’s linear growth of $O(N)$, the computational cost of round-robin grows quadratically at $O(N^2)$. When N=32, the computational cost of round-robin is already 8 times that of ArenaRL, making it a critical bottleneck in RL training.
>
> | Group Size N | ArenaRL Performance | Cost Ratio (Round-Robin / ArenaRL) |
> | :--- | :---: | :---: |
> | 8 | 32.5% | $2 \times$ |
> | 16 | 41.8% | $4 \times$ |
> | 32 | 46.9% | $8 \times$ |
>
> >**Q5:** Results of standard RL baselines equipped with process-aware rewards.
>
>
> **A5:** Thanks for your valuable comments. For fair comparison, we added the same multi-dimensional evaluation rubrics used in ArenaRL as process-aware rewards to standard GRPO and GSPO. The average performance on Open-Travel task is as follows:
>
> | Method | w/ Outcome-only Reward | w/ Process + Outcome Reward |
> | :--- | :---: | :---: |
> | GRPO | 16.4% | 18.9% |
> | GSPO| 17.2% | 18.3% |
> | ArenaRL | — | 41.8% |
>
> Adding process-aware scalar rewards slightly improves GRPO and GSPO, but their performance remains far below ArenaRL. This suggests that in open-ended tasks, pointwise scalar scoring still suffers from limited discriminative power even when process rewards are introduced. We will include this setting as an additional baseline in the revision.
>
> [1] Yang, et al. "HotpotQA: A dataset for diverse, explainable multi-hop question answering." EMNLP. 2018.
>
> [2] Du, et al. "Deepresearch bench: A comprehensive benchmark for deep research agents." arXiv:2506.11763.
>
> Thank you once again for your time and valuable feedback!

---

> > ### Author Rebuttal · Reviewer_jUVH · 2026-04-03
> >
> > Thank you for the detailed rebuttal.
> >
> > My main concern is the evaluation setup. This work proposes replacing pointwise scalar rewards with a tournament-based pairwise comparison scheme, which can be applied to any multi-step reasoning or tool-use datasets.
> > While the authors provide additional results on DeepResearch Bench, they still rely on GRPO with outcome rewards rather than stronger process-based baselines.
> >
> > The authors also do not include comparisons to tool-use or multi-step reasoning methods (e.g., SWIRL[1]), nor do they evaluate on additional standard datasets such as GSM8K and HotpotQA, which would provide a clearer and more controlled evaluation.
> >
> > Could the authors further explain why they focus on their proposed open-ended evaluation benchmarks and do not include more standard multi-step reasoning benchmarks or stronger baselines?
> >
> > [1] Goldie et al., Synthetic Data Generation & Multi-Step RL for Reasoning & Tool Use, CoLM 2025.

---

> > > ### Author Response · Authors · 2026-04-06
> > >
> > > Dear Reviewer jUVH:
> > >
> > > We sincerely appreciate your time and expertise in reviewing our work. We apologize that our previous explanation did not fully address your concerns and will now offer a clearer response to your questions.
> > >
> > > >**Q1:** Rationale for focusing on open-ended evaluation benchmarks
> > >
> > > **A1:** We would like to clarify that **ArenaRL is designed to address the unique evaluation challenges of open-ended tasks**. In such tasks, the correctness of intermediate steps or final outputs is often subjective and difficult to verify objectively. As a result, standard pointwise reward models suffer from discrimination collapse, producing noisy reward signals with low intra-group variance that fail to reflect true response quality. ArenaRL addresses this issue by replacing pointwise scoring with a tournament-based relative ranking mechanism, which provides more stable and discriminative training signals.
> > >
> > > In contrast, benchmarks such as GSM8K and HotpotQA have deterministic, verifiable answers. For these tasks, simple 0/1 outcome rewards based on answer matching are low-noise and empirically effective, and typically do not exhibit the discrimination collapse common in open-ended evaluation. Since our work focuses on open-ended, unverifiable task scenarios, our experiments primarily target open-ended benchmarks.
> > >
> > > >**Q2:** Evaluation on standard multi-hop reasoning benchmark HotpotQA
> > >
> > > **A2:** Thank you for the helpful suggestion. To evaluate the generality of ArenaRL on standard multi-hop reasoning tasks, we conducted additional experiments on HotpotQA and compared with SWIRL [1]. Following the SWIRL setup, we use Gemma 2-27B as the backbone model.
> > >
> > > Table 1. Performance comparison of ArenaRL and SWIRL on the HotpotQA benchmark
> > >
> > > |Method|Base Gemma 2-27b|SWIRL|ArenaRL|
> > > |:---|:---:|:---:|:---:|
> > > |HotpotQA|58.6|67.8|71.5|
> > >
> > > The results show that even on standard multi-hop reasoning tasks with verifiable answers, ArenaRL’s tournament ranking mechanism can still better distinguish high-quality reasoning trajectories and yield consistent performance improvements.
> > >
> > > >**Q3:** Comparison with stronger process-based baselines (e.g., SWIRL) on DeepResearch Bench
> > >
> > > **A3:** Thank you for pointing out the strong baseline SWIRL [1]. We reproduced SWIRL on DeepResearch Bench and used the same reward model (Qwen3-Max) as ArenaRL to provide step-level rewards. The results are shown below:
> > >
> > > Table 2. Performance comparison on DeepResearch Bench
> > >
> > > |Method|Overall|Comp.|Insight|Inst.|Read.|
> > > |:---|:---:|:---:|:---:|:---:|:---:|
> > > |GRPO|28.6|25.1|26.8|33.5|33.7|
> > > |SWIRL|32.3|29.2|30.8|37.0|35.2|
> > > |ArenaRL|39.2|38.4|35.9|43.7|42.5|
> > >
> > > While the process-based SWIRL improves over the outcome-only GRPO, it remains less effective than ArenaRL on complex open-ended tasks. This is because the correctness of intermediate steps in open-ended multi-hop reasoning tasks is inherently difficult to define effectively. Instead of assigning absolute scores to each step, ArenaRL relies on pairwise comparisons and ranking to extract stronger advantage signals, leading to better performance in open-ended multi-step reasoning.
> > >
> > > [1] Goldie et al., Synthetic Data Generation & Multi-Step RL for Reasoning & Tool Use, CoLM 2025.
> > >
> > > We deeply value your expertise and hope that our responses and the new results have addressed your concerns. We would be very grateful to know if you have any remaining questions or if there's anything that requires further clarification. If our rebuttal has successfully resolved your initial concerns, we would be deeply appreciative if you would consider re-evaluating our paper.
> > >
> > > Many thanks for your constructive comments, time, and patience.
> > >
> > > Best regards and thanks,
> > >
> > > The Authors

---

### Official Review · Reviewer_wHBD · 2026-03-13

**Soundness:** 2
**Presentation:** 3
**Significance:** 3
**Originality:** 2
**Overall Recommendation:** 5
**Confidence:** 3

**Summary:**

This paper introduces **ArenaRL**, a reinforcement learning framework designed to mitigate "discriminative collapse" in LLM agents for open-ended tasks. The authors argue that traditional pointwise reward models fail to distinguish subtle quality differences between trajectories due to low signal-to-noise ratios (SNR). To address this, ArenaRL adopts an **intra-group relative ranking** approach. To ensure computational feasibility, a **Seeded Single-Elimination Tournament** mechanism is proposed, reducing the complexity of pairwise comparisons from $O(N^2)$ to $O(N)$. The authors also contribute two new benchmarks: **Open-Travel** and **Open-DeepResearch**. Experimental results demonstrate that ArenaRL outperforms existing RL baselines like GRPO.

**Compliance With Llm Reviewing Policy:**

Affirmed.

**Final Justification:**

In my initial review, I gave a 3 (Weak Reject) primarily due to concerns regarding evaluation credibility (the GPT-4o anomaly), potential judge style bias, sensitivity to intransitive preference cycles, and a lack of empirical evidence for the "discriminative collapse" (SNR) hypothesis.

​The authors provided an exceptionally strong, evidence-based rebuttal that fully resolved all my concerns:
* Evaluation Credibility & Judge Bias: The detailed GPT-4o error analysis clarified the initial anomaly. More importantly, the new cross-judge training experiments (using GPT-4.1 and Claude-4-Sonnet) and t-SNE clustering convincingly demonstrated that ArenaRL optimizes for genuine task completion rather than overfitting to a specific judge's stylistic preference.
* ​Ranking Stability & SNR Evidence: The empirical measurement of preference cycles (only 1.16%) confirmed that the O(N) tournament is stable and not derailed by intransitive dynamics in this domain. Furthermore, the provided SNR tracking data solidly validated the core "discriminative collapse" motivation.

​Conclusion: The authors rigorously addressed all major technical red flags, significantly elevating the paper's soundness. The proposed O(N) tournament-based RL framework is a robust and practical contribution to the open-ended agent community. I am confidently raising my score to a 5 (Accept). I strongly encourage the authors to incorporate the rebuttal experiments (especially the cross-judge results and SNR tracking) into the camera-ready version.

**Key Questions For Authors:**

1. **GPT-4o Anomaly**: Could the authors provide a detailed error analysis for GPT-4o? Was the low win rate due to logical failure or format rejection by the Arena Judge?
2. **Cross-Judge Robustness**: Would the performance gains of ArenaRL persist if the RL-phase judge were replaced by a model from a different family (e.g., GPT-4o or Claude 3.5)?
3. **Intransitivity Analysis**: Have the authors analyzed the frequency of preference cycles in their benchmarks? How does the tournament handle non-transitive cases to ensure stable advantage signals?
4. **Stylistic Clusters**: Have the authors analyzed whether trajectories from different model families form distinct stylistic clusters, and whether this affects the consistency of the ranking?
5. **Empirical SNR**: Can the authors provide quantitative evidence (e.g., variance of $\sigma_{noise}$ vs $\sigma_{group}$) to support the claim of "discriminative collapse" in pointwise judges?

**Limitations:**

While the authors discuss the computational overhead of LLM-as-a-judge, they have not adequately addressed several critical limitations. First, the framework lacks an analysis of **non-transitive dynamics**; as discussed in *Lin et al. (2024)*, reducing $O(N^2)$ relationships to $O(N)$ rankings for efficiency may suppress "counter relationships" and introduce training instability. Second, the potential for **self-preference bias** (*Liu et al., 2025*) is not explored, especially given that the Arena Judge is from the same model family as the backbone. Lastly, the extreme underperformance of GPT-4o (2.6%) suggests that the benchmark might be sensitive to specific stylistic/formatting artifacts, which limits the claim of general reasoning improvement.

**Strengths And Weaknesses:**

## **Strengths**

1. **Motivation**: The paper identifies the SNR bottleneck in pointwise reward models for open-ended domains, providing a motivation for shifting towards ranking-based RL.
2. **Efficiency**: The proposed tournament mechanism offers a practical engineering solution to scale pairwise comparisons, which is traditionally a significant computational hurdle.
3. **Presentation**: The paper is clearly written, and the overall narrative—from the SNR motivation to the tournament approximation—is well-structured and easy to follow.
4. **Benchmarking**: The introduction of specialized agentic benchmarks adds value to the community by providing verifiable yet open-ended evaluation pipelines.

## **Weaknesses**

1. **Evaluation Credibility & GPT-4o Anomaly**: The reported results show a **win rate of only 2.6% for GPT-4o**. This is a significant outlier compared to most established agentic benchmarks. Such an extreme discrepancy raises concerns regarding the robustness of the evaluation protocol. If a decent generalist model fails so drastically, it may suggest that the benchmark is overly sensitive to specific stylistic or formatting patterns, rather than objective task-solving capability. **A simple sanity check using an independent judge would be necessary to determine whether the ranking remains consistent.**
2. **Judge Bias & Stylistic Adaptation (Ref: Liu et al., 2025)**: As discussed in *Liu et al. (2025)*, open-ended LLM-based evaluations can exhibit self-preference bias and other systematic artifacts. Since ArenaRL relies on a single-family judge (Qwen3-Max) during the RL training phase, it remains unclear whether the learned policy reflects improved task reasoning or is merely **adapting to the judge's implicit stylistic preferences**. Without cross-family judge validation, the generalizability of the reward signal is difficult to assess.
**3. Sensitivity to Intransitive Preferences and Limitations of Complexity Reduction (Ref: Lin et al., 2024)**:
A critical technical concern lies in the potential oversimplification of trajectory relationships. While the authors propose a **Seeded Tournament** to reduce the complexity of pairwise comparisons from $O(N^2)$ to $O(N)$, they do not adequately address the inherent **non-transitive dynamics** (e.g., preference cycles like A > B > C > A) common in multi-dimensional open-ended tasks.
Prior research on competitive systems (e.g., **Lin et al., 2024**) specifically explores the challenge of reducing $O(N^2)$ matchups to $O(N)$ ratings. Notably, Lin et al. demonstrate that even when computational complexity is optimized, **purely scalar ratings fail to capture essential "counter relationships" (intransitivity)**, necessitating further techniques such as **clustering or latent style decomposition** to maintain evaluation integrity.
By forcing a linear ranking through a tournament structure for advantage estimation, ArenaRL may be suppressing these cycles into noisy or unstable gradient signals. The authors should discuss whether their $O(N)$ approximation risks losing these critical non-transitive nuances and whether a more sophisticated decomposition of "style clusters"—as suggested in related competitive balance literature—is required to ensure training stability.
4. **Incremental Technical Originality**: Conceptually, the transition from pointwise rewards to relative group-based ranking is well-explored in existing literature (e.g., GRPO, GSPO). While the **Seeded Single-Elimination Tournament** is a novel application to improve computational efficiency ($O(N)$ complexity), this contribution is primarily an **algorithmic approximation** of pairwise preferences rather than a fundamental shift in RL training objectives. The significance of this work thus rests heavily on the robustness of the tournament results, which remains a concern given the potential for judge bias and intransitivity.

* *Liu, S., et al. (2025). Re-evaluating Open-ended Evaluation of Large Language Models. ICLR.*
* *Lin, C. C., et al. (2024). Identifying and Clustering Counter Relationships of Team Compositions in PvP Games for Efficient Balance Analysis. TMLR.*

---

> ### Author Rebuttal · Authors · 2026-03-31
>
> Dear Reviewer wHBD:
>
> We sincerely thank you for the valuable comments! Your comments are crucial for us to improve our work. We will address your concerns point by point.
>
> >**Q1:** Detailed error analysis of GPT‑4o.
>
> **A1:** Thanks for your suggestion. We analyzed GPT‑4o’s generation samples and found a tendency toward premature termination before retrieving all required information for a query. Specifically, 23.2% of cases skip tool usage entirely to output a final answer, 11.2% involve tool execution failures, 16.5% use invalid tool parameters, and 5.3% perform redundant searches. This leads the model often produces low-quality outputs with incomplete information. This aligns with prior work [1], where GPT‑4 achieved only 0.6% success rate in fully satisfying all constraints on the a similar travel planning task.
>
> To check for Arena judge fairness, we also used GPT‑4.1 to evaluate GPT-4o's win rate against the Claude-4-Sonnet baseline. While its mean win rate slightly increases (from 2.6% to 4.6%), it remains low overall.
> |Model|Direction|Search|Compare|1-Day|M-Day|Mean|
> |:---|:---:|:---:|:---:|:---:|:---:|:---:|
> |GPT-4o|3.7|9.1|5.3|2.8|1.9|4.6|
> >**Q2:** Judge bias and performance with different judge model families.
>
> **A2:** As shown in Sec. 5.2 of our paper, our benchmark uses a dual-judge setup. During training, Qwen3‑Max serves as the reward model; during evaluation, Qwen3‑Max and Claude‑4‑Sonnet independently judge the outputs, and the final score is the average win rate. And in Fig. 4(b), LLM evaluation achieves 73.9% agreement with human annotations, indicating strong human alignment.
>
> To test cross‑judge robustness, we evaluate Open‑Travel task using GPT‑4.1 and Claude‑4‑Sonnet as reward models. ArenaRL achieves consistent gains under all three judges, indicating that the improvement is not tied to specific model family.
> |Model|Direction|Search|Compare|1-Day|M-Day|Mean|
> |:---|:---:|:---:|:---:|:---:|:---:|:---:|
> |SFT|10.6|29.7|14.1|20.4|7.1|16.4|
> |GPT-4.1|27.4|57.3|32.5|53.8|18.4|37.9|
> |Claude-4-Sonnet|35.2|64.3|34.6|57.4|24.6|43.2|
> |Qwen3-Max|32.1|66.1|31.7|58.0|21.0|41.8|
> >**Q3:** Frequency of preference cycles and how ArenaRL mitigates non-transitivity?
>
> **A3:** Thanks for pointing out the related work. We conducted round‑robin comparisons within 10 trajectory groups (16 responses per group), constructing an N×N preference matrix for each group and counting cycles of length three or more. The frequency is only 1.16%.
>
> ArenaRL mitigates non-transitivity through three key designs: **(1) Multi‑dimensional rubrics:** Scoring across seven dimensions prevents non-transitivity in any single metric from dominating. **(2) Seed‑based pre‑ranking:** Using a deterministic greedy‑decoded trajectory as a shared anchor and compares it against all other trajectories to provide a reliable prior ranking. **(3) Cumulative scoring:** final rankings are determined by accumulated scores across multiple matches, smoothing occasional non‑transitive outcomes.
>
> >**Q4:** Style clustering and ranking consistency across model families
>
> **A4:** Under the same system prompt, we generated trajectories for 20 queries using GPT‑4.1, Claude‑4‑Sonnet, and Qwen3‑Max. Features were extracted using Qwen3‑Embedding-8B and visualized with t‑SNE. The embeddings cluster primarily by query content rather than model family, with no clear family‑specific style clusters.
>
> For ranking consistency, we independently ranked 10 trajectory groups using the three models yielded high Spearman correlation ρ: (1) GPT‑4.1 vs Claude‑4‑Sonnet: ρ = 0.938; (2) GPT‑4.1 vs Qwen3‑Max: ρ = 0.927; (3) Claude‑4‑Sonnet vs Qwen3‑Max: ρ = 0.951. This confirms rankings are driven by task completion quality rather than the stylistic preferences of specific judges.
>
> >**Q5:** Quantitative evidence of discriminative collapse
>
> **A5:** We tracked the variance of $\delta_{group}$ and $\delta_{noise}$ during RL training to compute the signal-to-noise ratio (SNR).
> |Step|$\delta_{group}$|$\delta_{noise}$|SNR|
> |:---|:---:|:---:|:---:|
> |0|0.046|0.024|1.92|
> |20|0.038|0.026|1.46|
> |40|0.031|0.023|1.35|
> |60|0.034|0.024|1.42|
> |80|0.029|0.023|1.26|
>
> Due to the cold‑start stage, model already has strong capability, with the SNR at step 0 is only 1.92. It further decreases and stabilizes around 1.3, providing clear quantitative evidence of discriminative collapse:
>
> >**Q6:** Technical originality compared with GRPO/GSPO
>
> **A6:** We summarize the key differences of ArenaRL over GRPO/GSPO:
>
> |Method|Reward Acquisition|Advantage|Evaluation Granularity|Complexity|Open-Travel Performance|Open-DeepResearch Performance|
> |:---|:---:|:---:|:---:|:---:|:---:|:---:|
> |GRPO/GSPO|Pointwise reward|Group-normalized rewards|Outcome-only|$O(N)$|16.4/17.2|25.2/25.2|
> |ArenaRL|Pairwise preferences|Preference ranking|Process+Outcome|$O(N)$|41.8|64.3|
>
>
> [1] Xie, et al. TravelPlanner: A Benchmark for Real-World Planning with Language Agents, ICML 2024
>
> Thank you once again for your time and valuable feedback!

---

> > ### Author Rebuttal · Reviewer_wHBD · 2026-04-02
> >
> > ​I would like to commend the authors for an exceptionally thorough and evidence-based rebuttal. The authors have systematically addressed all my major concerns with rigorous new experiments.
> >
> > ​Specifically, the detailed failure analysis of GPT-4o, supported by prior literature (Xie et al., 2024), clarifies the initial anomaly. Most importantly, the new cross-judge training experiments (using GPT-4.1 and Claude-4-Sonnet) and the t-SNE style clustering analysis effectively dispel my concerns regarding "self-preference bias" and "stylistic overfitting."
> >
> > Furthermore, the empirical quantification of preference cycles (only 1.16%) and the SNR tracking provide the missing solid foundation for the paper's core claims.
> > ​Given the significant improvements in empirical validation and the practical utility of the proposed O(N) tournament framework for open-ended RL, I am raising my score from 3 to 5 (Accept). The paper now presents a highly robust evaluation and makes a solid contribution to the community.

---

> > > ### Author Response · Authors · 2026-04-02
> > >
> > > Dear Reviewer wHBD:
> > >
> > > We sincerely appreciate your insightful review. Your positive feedback and support for our work are truly encouraging to us.
> > >
> > > We are pleased to know that our rebuttal has fully addressed your concerns. We are deeply grateful for the professional guidance you provided throughout the review process.
> > >
> > > Many thanks for your constructive comments, time, and patience.
> > >
> > > Best regards and thanks,
> > >
> > > The Authors

---

### Decision · Program_Chairs · 2026-04-30

**Decision:**

Accept (regular)

**Comment:**

This paper proposes ArenaRL, which replaces pointwise scalar rewards with tournament-based relative ranking for reinforcement learning in open-ended agent tasks. It also introduces two new benchmarks for training and evaluating open-ended agents. Three reviewers vote for accept and one reviewer suggest for week accept. The main issues raised in review concerned evaluation robustness, judge bias, baseline strength, and tournament design. The authors addressed these concerns in the rebuttal with additional experiments and analysis, and the reviewer discussion after the rebuttal was broadly positive. I recommend acceptance.